# Myoferlin is a component of late-stage vRNP trafficking vesicles for enveloped RNA viruses

Stefano Bonazza ⬡ , Hannah L. Turkington, Swathi Sukumar ⬡ , Emily Peate, Hannah L. Coutts, Joshua J. Montgomery, Courtney Hawthorn, Erin M. P. E. Getty, Olivier Touzelet, Judit Barabas, Ultan F. Power ⬡ & David G. Courtney ⬡ ✉

The Rab11 endosomal recycling pathway is exploited by important respiratory RNA viruses such as IAV and RSV, aiding viral egress from the apical surface of polarized epithelial cells. Late in infection, Rab11-containing vesicles specifically transport viral ribonucleoprotein (vRNP) complexes towards the cell surface before packaging and budding. Rather than employing traditional Rab11-positive recycling endosomes, virus-infected cells generate remodelled Rab11-containing vesicles, as observed during IAV infection. Besides Rab11, no other conserved host co-factors have been identified among these various vRNP trafficking vesicles. Here we discover and confirm myoferlin's association with IAV vRNPs in the cytoplasm and colocalisation with Rab11 during late stages of infection. We also find that this role is conserved in late-stage vRNP trafficking of other viruses, including RSV and SeV. Myoferlin likely recruits the EHD family of proteins, which are involved in endosomal biogenesis, to these unique vRNP trafficking endosomes, highlighting myoferlin's pivotal role in viral replication.

Cumulatively, respiratory viral pathogens including influenza A virus (IAV) and respiratory syncytial virus (RSV), represent a major annual global burden, resulting in the estimated death of around 5 million people every year[1]. While these two highlighted pathogens are both RNA viruses infecting and transmitting from the human respiratory tract, biologically they are clearly distinct in how they replicate within host cells. For instance, IAV has a segmented negative-sense genome and replicates in the nucleus, while RSV has a single-segment negative-sense genome and replicates in the cytoplasm. Despite these differences, a small number of cellular systems have been found to be widely exploited by these and several other RNA viruses. Rab11-mediated vesicular trafficking is one such system[2–6].

The Rab11-containing endosomal recycling pathway constitutes a major cellular trafficking system widely remodelled by enveloped respiratory RNA viruses[7,8]. This is true for human pathogens, including IAV and RSV, as well as mammalian respiratory pathogens, such as Sendai virus (SeV)[9]. It has also been speculated that coronaviruses exploit this pathway during egress[10]. At late stages of infection of each virus, Rab11-containing vesicles are exploited to traffic viral ribonucleoprotein (vRNP) complexes towards the apical surface so that virions are dispersed into the respiratory airways[3,11–13]. A better understanding of this cellular trafficking system in both the presence and absence of infection could reveal important, novel insights broadly applicable to many virus families. In humans, Rab11 is present as three very similar isoforms encoded by three different genes: Rab11, Rab11b and Rab25. The slight differences between isoforms are understudied, in particular in the context of IAV infection, with the majority of work on Rab11. Hence, we will refer to Rab11 isoforms as a whole throughout the manuscript. Additionally, it is now clear that it is not the traditional Rab11 recycling endosomes that are simply hijacked and repurposed for vRNP trafficking[7,14]. Instead, uniquely remodelled Rab11-containing irregularly coated vesicles (ICV) appear to form in virus-infected cells, at least for IAV vRNP trafficking, with the involvement of structures from the endoplasmic reticulum[7,15,16]. Aside from

Wellcome-Wolfson Institute for Experimental Medicine, Queen's University Belfast, Belfast, UK. ✉e-mail: david.courtney@qub.ac.uk

the common utilisation of Rab11, very little is known of the other co-factors that are likely conserved among these vRNP trafficking vesicles that arise during various viral infections. Therefore, the identification of a co-factor that, during infection by various viruses, is found to associate on newly formed Rab11-containing vRNP trafficking vesicles would be critical in furthering our understanding of generalised directional vRNP egress.

To this end, an immunoprecipitation-mass spectrometry approach was employed using an epitope-tagged influenza polymerase protein at a late stage of infection to uncover host factors of the viral trafficking endosome, aside from Rab11, that are essential for influenza virus replication. After analysis of these data, the trans-membrane protein myoferlin, MYOF, was found to clearly associate with IAV vRNPs in the cytoplasm at late stages of infection and strongly colocalised with Rab11 in both the presence and absence of infection. Additionally, the replication capabilities of clinical and lab isolates of IAV were strongly reduced after siRNA-mediated knockdown of MYOF expression.

Further investigations were undertaken to establish whether this strong association between MYOF and Rab11-trafficking vesicles persisted on vRNP trafficking endosomes of other respiratory viruses that traffic with directionality to the apical surface of airway epithelial cells. RSV and SeV were both found to be sensitive to MYOF levels in the cell, in the same manner as observed for IAV, while imaging confirmed a strong association between MYOF, Rab11 and vRNPs on trafficking endosomes, recapitulating the observation made during IAV vRNP trafficking. Finally, it has previously been shown that MYOF can associate with EHD proteins[17], which are involved in the late-stage maturation and budding of a subset of endosomes[18,19]. Therefore, the possibility remained that MYOF in these vesicles recruits EHD proteins as an important step in vesicular biogenesis. On investigation, it was revealed that EHD1 and EHD2 are clearly present within these IAV vRNP trafficking endosomes. Herein, we describe the discovery of MYOF as an essential factor for vRNP trafficking vesicle biogenesis and that it likely mediates the recruitment of EHD proteins to complete vesicular remodelling.

## Results

### Identification of the PA interactome

IAV vRNPs are known to traffic on Rab11-containing vesicles towards the plasma membrane at late stages of infection. This work aimed to uncover potential host factors, in addition to Rab11, that are essential for this late stage of virus replication. A reporter strain of influenza A/WSN/33 encoding a FLAG-tagged PA subunit (henceforth referred to as WSN_PA-FLAG) was designed and rescued and was used to selectively immunoprecipitate (IP) the PA interactome as a proxy of the vRNP. Paraformaldehyde (PFA) crosslinking was employed to capture even transient and long-range interactors, while also allowing for the use of more stringent wash steps. Moreover, the IP was carried out in the presence of RNases to focus this search exclusively on the protein-protein interactome of the vRNP (Fig. 1A). The specificity and stringency of the IP protocol were assayed by western blotting and silver staining of the input samples and the IP fractions. As a baseline and negative control for the IP, the wild-type influenza A/WSN/33 (WSN_WT) was used. A robust enrichment of PA-FLAG, but not wild-type PA, was observed in IP samples, as well as the absence of actin, which is not a known interactor of the vRNP (Fig. 1B). Silver staining demonstrated a clear enrichment in overall proteins after FLAG IP from cells infected with WSN_PA-FLAG over WSN_WT (Fig. 1C).

The recombinant WSN_PA-FLAG replicates at a lower rate compared to wild-type, displaying a slightly delayed infection cycle (Fig. S1A). On performing IF on A549 cells we noted that 6 and 16 h post infection (hpi) represented 2 distinct stages in the infectious cycle for the PA-Flag tagged WSN virus (Fig. S1B). At 6 hpi PA was predominantly localised to the nucleus, representing an early stage of active RNA

synthesis, while at 16 hpi PA was predominantly localised to the cytoplasm, representing a late stage of vRNP nuclear export and trafficking. Therefore, the experiment was performed at both 6 and 16 hpi, modelling early and late infection, respectively, thereby providing a more exhaustive picture of the vRNP interactome throughout the viral replication cycle. The early interactome produced fewer significant hits (Fig. 1D and Supplementary Data 1), perhaps due to the comparatively low expression of viral proteins. In contrast, the late interactome contained proteins involved in translation, protein folding and intracellular protein transport (Figs. 1E and S1; Supplementary Data 2). When comparing proteins enriched in both early and late timepoints, a strong enrichment of the influenza polymerase components PB2, PB1, PA and NP was observed as expected (Fig. 1F and Supplementary Data 3), in addition to host factors including MYOF, CCT8 and CACYBP. Finally, when directly comparing the early to late interactome to assess the dynamics of some of these host interactors, 10 proteins were significantly enriched at 6 hpi vs. 16 hpi (Fig. 1G, left and Supplementary Data 4), while a large set of 255 proteins were significantly enriched at 16 hpi vs. 6 hpi (Fig. 1G, right and Supplementary Data 4). Interestingly, upon STRING analysis of a subset (enrichment >3 fold) of these 16 hpi enriched proteins, a clustering of proteins associated with the nucleocytoplasmic transport complex was observed, as expected for mature vRNPs shuttling from the nucleus at late stages of infection (Fig. S2, circled red). More intriguingly, we found that 14 of these strongly enriched PA-interacting proteins (Fig. S2, circled blue) were previously identified as host proteins reliably packaged into WSN virions[20]. In fact, these 14 proteins make up 40% of the 35 total host proteins identified in this study, and constitute a reliable indicator that this 16 hpi interactome is indeed capturing late-stage vRNP interacting proteins.

### MYOF associates with vRNPs at late stages of infection

These interactomes constitute an important snapshot of host-pathogen interactions. In particular, the late interactome comprised a collection of putative host trafficking determinants. From this dataset, the six most significantly enriched hits were investigated via siRNA knockdown, followed by infection, and quantification of viral replication. A549 cells were transfected with siRNAs targeting CACYBP, CCT8, EEIF1G, MYOF, PAICS, and PFDN5, as well as a non-targeting control (NSC) and PABPC1 as a negative control[21]. After 48 h cells were infected with MOI 3 wild-type WSN for 8 h, then supernatants were harvested and titrated. Of the candidate genes, only depletion of MYOF or PFDN5 produced a significant reduction in viral replication (Fig. 2A). Among the candidates that strongly affected IAV titres, MYOF was chosen as a promising candidate for mediating viral trafficking, as previous work linked it to membrane dynamics and endocytosis[22].

The effect of myoferlin knockdown on IAV was not a strain-specific phenotype, since a similar, if not more pronounced, reduction in replication was observed for the post-pandemic H1N1 A/Brisbane/02/2018 (Brisbane18), and the H3N2 strain A/Shanghai/24/90 (Shanghai90) (Fig. 3C) in cells transfected with siRNA at a concentration of 4 nM for 48 h prior an 8 h infection.

Figure 2B–D shows the effect of MYOF knockdown on viral replication, gene expression and viral protein levels. Cells were transfected with either NSC (4 nM) or MYOF siRNA (4 nM) for 48 h prior to infection with MOI 3 WSN for 8 h. The siRNA concentration was chosen as it did not appreciably affect cell viability (Fig. S3A) but did consistently reduce viral titres (Fig. S3B). Supernatants from these cells were then harvested and titrated (Fig. 2C), while the cells were lysed for RNA extraction and qPCR (Fig. 2D). Treatment with 4 nM of the MYOF siRNA successfully depleted the amount of MYOF mRNA to ~28% of the control, resulting in a ~ 5-fold reduction in infectious particles released. Importantly, this is not accompanied by a drop in either mRNA or vRNA synthesis, suggesting a clear involvement in post-replication steps, as has been demonstrated previously upon loss of Rab11

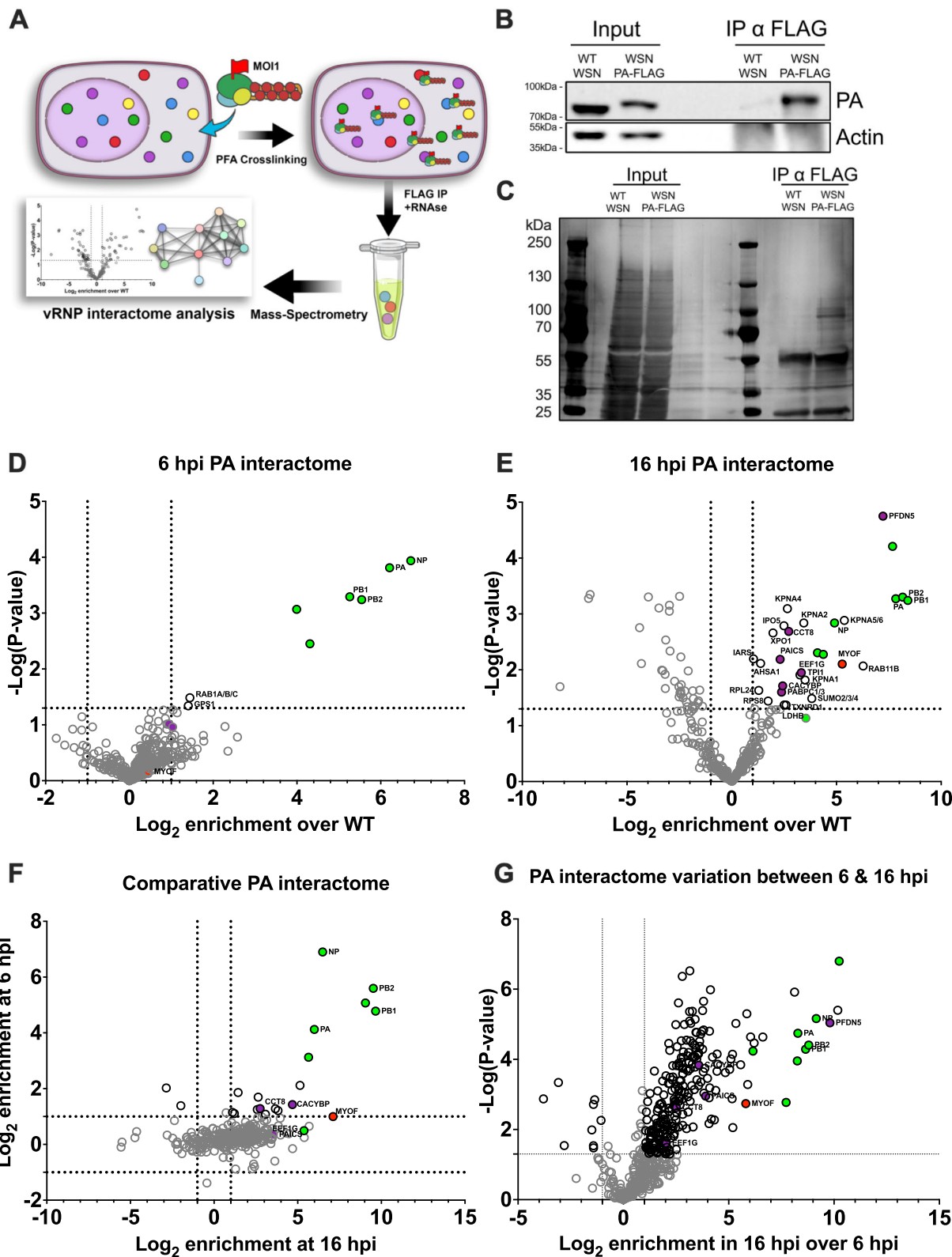

expression[23]. We then employed a known MYOF small molecule inhibitor, WJ460, to assess its effect on IAV replication. Closely mimicking our siRNA data, cells treated with WJ460 significantly reduced WSN titres at 8 hpi (Fig. S3D) while MYOF, NP and NS1 protein and MYOF, NP mRNA and NP vRNA transcript levels remained unaffected (Fig. S3E, F), further supporting our conclusions that MYOF is an important host factor in late stage IAV replication.

Immunofluorescence was then employed to assess myoferlin's potential as a trafficking factor for IAV vRNPs. A549 or the immortalised bronchial epithelial cell line BEAS-2Bs were infected with either WSN_WT for 8 h or the H1N1 A/California/07/2009 (Cal09) for 16 h, then fixed and stained for MYOF and NP proteins, as a proxy for vRNPs. As shown in Fig. 2E and quantified in Fig. 2F by Pairwise Pearson's Correlation Coefficients (PCC), myoferlin colocalised extensively with

**Fig. 1 | Identification of the PA interactome. A** Schematic of the immunoprecipitation approach used. A549 cells were infected with the IAV WSN strain encoding a FLAG-tagged PA polymerase subunit for either 16 or 6 h at an MOI of 1, then crosslinked with PFA and harvested. The PA interactome was isolated by FLAG immunoprecipitation in the presence of RNases to isolate exclusively protein-mediated interactions and subjected to mass spectrometry. **B** Representative Western blotting of the 16 hpi IP samples analysed. **C** Representative silver staining of the same IP samples. Uncropped images are shown in Fig. S7. **D** Volcano plot of the early interactome (6 hpi). Viral proteins are represented in green, while purple indicates the hits that were followed up on. Lastly, red indicates myoferlin.

**E** Volcano plot of the late interactome (16 hpi). **F** Correlation plot of the early and late interactomes. The two datasets were analysed concurrently, and for each entry, an enrichment value for both 6 h and 16 h was determined. Proteins in the top right quadrant are enriched in both datasets, the ones in the top left are only enriched in the early. **G** Volcano plot representing the differential enrichment of proteins between the early FLAG samples and late FLAG samples. Each dataset is derived from three independent biological replicates. For all pairwise comparisons, we used an unpaired two-sided (two-tailed) Student's $t$ test in Perseus and accounted for multiple testing with a 5% FDR, with results displayed as −log10 $p$ values in the volcano plots.

NP in the cytoplasm, particularly in the juxtanuclear region, as shown in the magnified images (inset). Myoferlin's colocalisation with the vRNPs was further assayed by single-molecule inexpensive fluorescent in situ hybridisation (smiFISH). A549 cells were infected at MOI 5 with WSN_WT for 8 h, fixed and stained for NA vRNA (Seg 6), NP protein, and MYOF (Fig. 2G). Correlation analysis (Fig. 2H) confirmed the strong colocalisation between MYOF and vRNPs.

The significant effect of myoferlin depletion on IAV replication, its extensive colocalisation with vRNPs during late-stage infection, and particularly its intracellular localisation appearing comparable to that of the canonical endocytic recycling machinery, provided a rationale for a deeper study of myoferlin's role in the recycling pathway alongside Rab11.

### Rab11 and MYOF work in tandem to coordinate slow endocytic recycling

The potential relationship between myoferlin and Rab11 was initially investigated in the absence of infection, to better ascertain the role myoferlin may play in cellular recycling under normal growth conditions. First, it was observed that MYOF and Rab11 strongly colocalised by immunofluorescence in an A549_Rab11-mCherry reporter cell line, as shown in Fig. 3A and quantified in Fig. 3B. Colocalisation was not limited to the prominent juxtanuclear accumulation but extended to the smaller foci at the cellular periphery (as shown in the magnification), suggesting an overlap throughout trafficking.

The extensive colocalisation suggested the possibility of a functional relationship between MYOF and Rab11 in the orchestration of the endocytic recycling compartment (ERC). Therefore, the effect of the ablation of one protein on the localisation of the other was next explored. The siRNAs successfully greatly reduced myoferlin and Rab11 protein levels by IF (Fig. 3C). Knockdown of Rab11 profoundly affected MYOF localisation, with its signal aggregating at a juxtanuclear spot in the absence of the GTPase (Fig. 3C). Similarly, myoferlin depletion led to a pronounced remodelling of the Rab11 network, resulting in larger juxtanuclear accumulations of Rab11 (Fig. 3C). These aggregating phenotypes were confirmed by quantifying the area of the largest cluster of signals in the cytoplasm of each cell for each condition (Fig. 3D, E). To determine whether this relocalisation was specific to Rab11 recycling endosomes and not a general phenotype of endosomes in the absence of MYOF, we also imaged EEA1, Rab7 and LAMP1 localisation, representing early endosomes, late endosome and lysosomes, respectively (Fig. S4A). Here we did not observe any clear differences in localisation of any of these markers upon loss of MYOF expression compared to control cells. As Rab11 localisation is perturbed in the absence of MYOF, it stands to reason that Rab11-dependent recycling would be altered as well. Indeed, cells depleted for MYOF showed a marked redistribution of Atto647-tagged transferrin in a transferrin uptake assay (Fig. S4B, C). After 30 min of transferrin uptake, most of the signal in MYOF-depleted cells accumulated in large foci near the nuclear membrane, while in control cells it was dispersed in smaller distinct puncta throughout the cytoplasm. This was also recapitulated by Rab11 knockdown. Indeed, this diminishment in cellular recycling has been observed previously in MYOF null myoblasts[17]. Here authors found that the loss of myoferlin resulted

in delayed recycling to the cytoplasmic membrane, though no corresponding investigations into Rab11 loss or its association with MYOF was performed.

Taken together, these data suggest a role for myoferlin in Rab11-mediated recycling in uninfected cells, thereby confirming (at least in this A549 lung-cell model) the intimate relationship between the two. However, during IAV infection the Rab11+ ERC is severely remodelled, affecting the localisation, composition, and recycling function of the whole vesicular network[7,15]. Therefore, the presence of MYOF in these distinct virus-induced vesicles was next explored.

### MYOF is specifically retained in influenza A vRNP-trafficking vesicles

IAV infection profoundly affects the ERC, particularly during late-stage infection. It remodels the structure and composition of Rab11+ vesicles. So far, we have shown that Myoferlin and Rab11 work in concert to ensure proper recycling in healthy cells, but nothing is known about myoferlin's fate upon infection with IAV. To adequately observe the remodelling of the ERC upon infection, the reporter A549_Rab11-mCherry cell line was infected for 8 h with WSN_WT and imaged for NP, mCherry and MYOF. Compared to mock-infected cells, infected cells showed a marked redistribution of the Rab11-mCherry signal, displaying several enlarged foci strongly colocalising with NP, and importantly MYOF (Fig. 4A, B).

Colocalisation analysis on its own would not be sufficient to prove MYOF's inclusion in irregularly coated vesicles (ICVs), due to the small and dispersed nature of these vesicles. To bypass this issue, infected cells were treated at 6hpi with nucleozin, an NP-crosslinking drug[24], to induce the selective aggregation of vRNP-associated vesicles. Thus, ICVs and all their constituents would coalesce into easily detectable foci. As shown in Fig. 4C, the drug had no effect in the absence of infection but caused the formation of large inclusions in infected cells. As expected, these inclusions were positive for NP and Rab11. Interestingly, they were also MYOF-positive. Importantly, nucleozin addition did not alter the localisation of other subcellular compartment markers such as Rab7 (late endosomes) or LAMP1 (lysosomes) (Fig. 4D), indicating a specific effect on vRNP trafficking vesicles only. Moreover, these observations were confirmed in BEAS-2B (Fig. 4E). PCCs were calculated for single cells in the presence or absence of the drug, showing an increased colocalisation of NP and MYOF, NP and Rab11, and Rab11 and MYOF, but not for the negative controls Rab7 and LAMP1, following aggregation (Fig. 4F). This established the presence of MYOF in ICVs, providing evidence of its likely functional partnership with Rab11 even during infection.

### Respiratory viruses commonly utilise Rab11-MYOF vesicles at late stages of infection

The trafficking of vRNP complexes of respiratory viruses other than IAV, including RSV and SeV, is known to depend on Rab11-containing vesicles. To investigate whether replication of these viruses is also dependent on myoferlin, we performed infections in siRNA treated A549 cells followed by titration of released virus and RNA quantification, similar to Fig. 2C, D. siMYOF treated cells infected with RSV_mKate2 at an MOI 2 for 24 h yielded a significant >10-fold

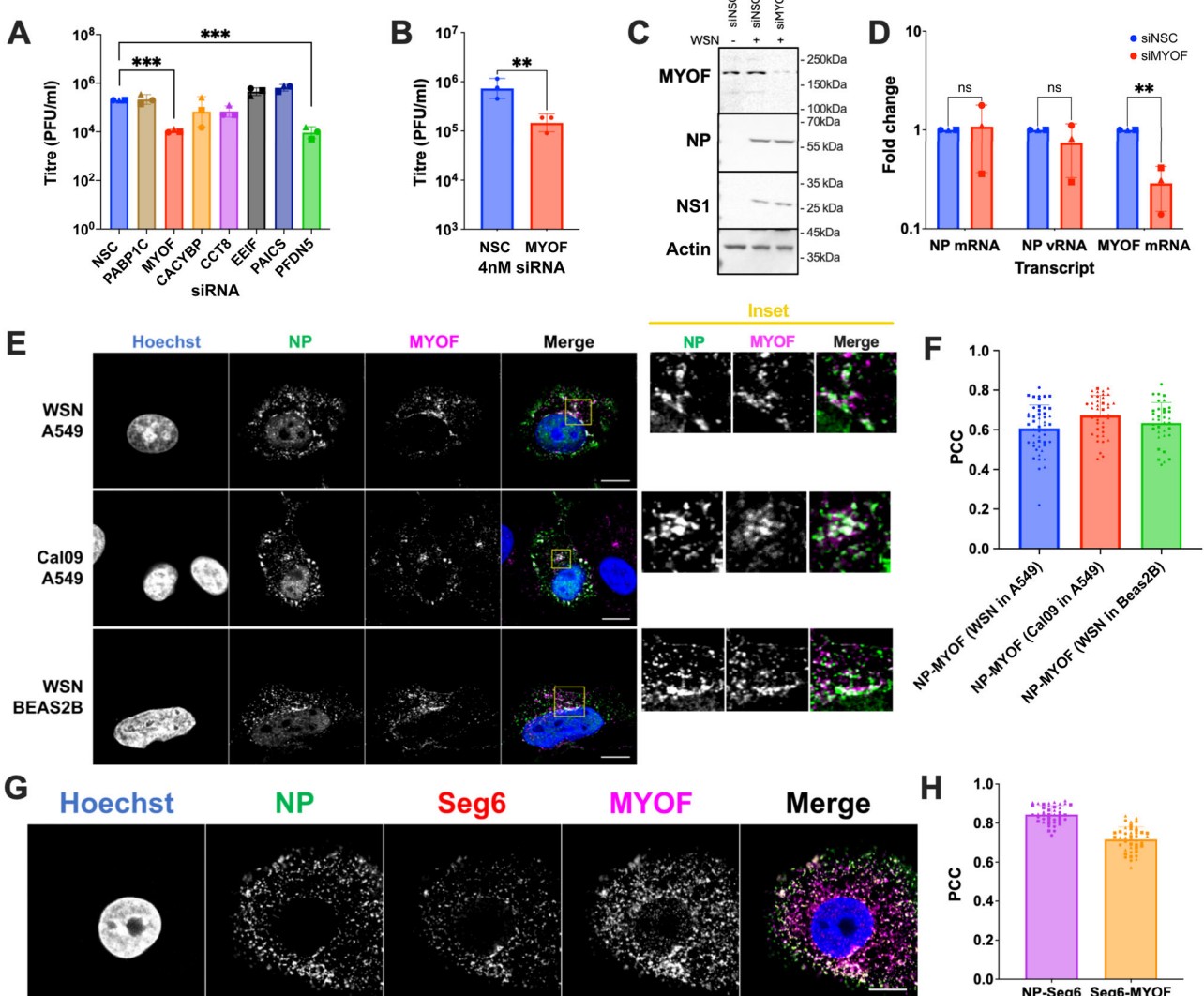

**Fig. 2 | MYOF associates with vRNPs at late stages of infection. A** Titres of supernatants from siRNA transfected A549 cells that were subsequently infected with MOI 3 with WSN for 8 h. 10 nM of siRNAs were used targeting either a non-specific control (NSC), a negative control (PABPC1) or a candidate hit from the interactome analysis (CACYBP, CCT8, EEIF1G, MYOF, PAICS, PFDN5). **B** Titres from supernatants of siRNA-transfected A549 cells infected at MOI 5 with WSN for 8 h. Infected cells were then washed and either protein (**C**) or RNA (**D**) was harvested with protein levels visualised by Western or relative transcripts levels quantified by qPCR. Uncropped images are shown in Fig. S8. **E** A549 cells (top and middle row) or BEAS-2B cells (bottom row) were infected with MOI 5 WSN for 8 h (top and bottom row) or Cal09 MOI 1 for 16 h (middle row), then fixed and stained for NP (green), and MYOF (magenta), and counterstained with Hoechst. The perinuclear accumulation in the yellow inset is magnified on the right. Scale bar 10 μm. **F** Pearson's correlation coefficient quantification of the colocalisation between MYOF and NP in single cells in the indicated conditions. Across 3 biological replicates correlation coefficients for 49, 41 and 35 individual cells for NP-MYOF in WSN in A549, Cal09 in A549 and WSN in Beas2B conditions, respectively. **G** A549 cells were infected for 8 h with MOI 5 WSN, then fixed and processed for smiFISH to stain segment 6 (NA, red), followed by immunofluorescence to visualise NP (green) and MYOF (magenta). Scale bar 10 μm. **H** Pearson's correlation coefficient quantification of the pairwise colocalisation between NP, MYOF, and segment 6, in 43 single cells across 3 biological replicates. In **F**, **H** different shaped points indicate individual cells from different replicates. For **A**, **B** data were analysed by ordinary one-way ANOVA followed by a two-sided Dunnett's multiple-comparisons test (each siRNA vs. NSC), and for **D** by two-way ANOVA with a two-sided Dunnett's multiple-comparisons test, with $p$ values adjusted to control the family-wise error rate at $\alpha = 0.05$ with ** = $p < 0.01$ and *** = $p < 0.001$. All experiments consist of 3 biological replicates, with the bar representing the mean of the replicates and error bands representing standard deviation.

reduction in titres (Fig. 5A) when compared to siNSC treated cells, while viral mRNA levels were unaffected (Fig. 5B). The same was true for siRNA-treated A549 cells infected with SeV_eGFP where a > 5-fold decrease in titres was measured (Fig. 5C) with no corresponding effect on viral mRNA (Fig. 5D). smiFISH imaging was performed to visualise RSV or SeV vRNAs during infection of A549 cells. These data revealed a distinct colocalisation between the genomic RNAs and MYOF (Fig. 5E, F). A549_Rab11-mCherry cells were infected with RSV_BT2a (a low passage clinical isolate), while A549 cells were infected with SeV_eGFP. After 24 h the samples were stained for their respective vRNAs and

MYOF. For both viruses, the most abundant colocalisation occurred within irregularly shaped vRNA aggregates. The same phenomenon was also observed for the reporter RSV strain (RSV_mKate2, Fig. S6A), while A549_Rab11-mCherry cells were also infected with SeV_eGFP demonstrating sustained colocalisation between MYOF and Rab11 upon infection (Fig. S6B, C). This is consistent with a model in which the Rab11-MYOF pathway is used specifically to transport mature genomes towards sites of virion budding. Interestingly, for RSV-infected cells, infection seemed to induce a strong re-localisation of the MYOF signal, indicative of ERC remodelling. Some of this signal

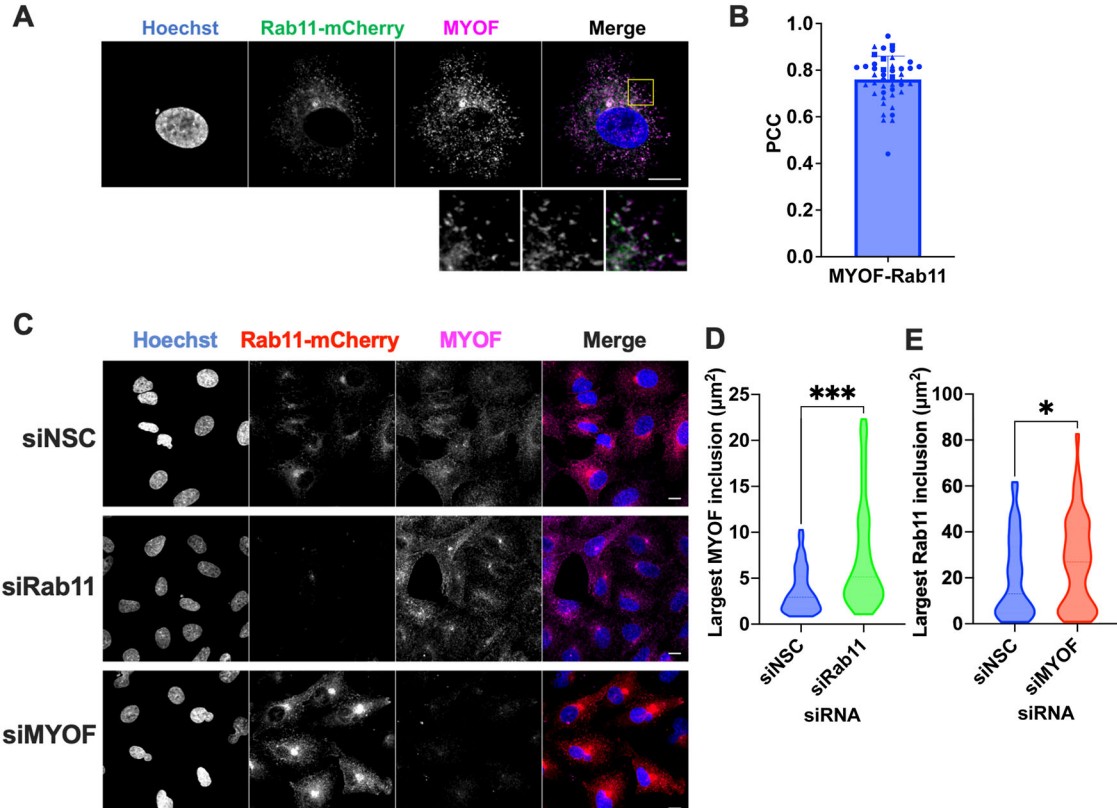

**Fig. 3 | Rab11 and MYOF work in tandem to coordinate slow endocytic recycling. A** A549_Rab11-mCherry cells were stained for MYOF and imaged. The yellow inset is shown magnified below. Scale bar 10 μm. **B** Pearson's correlation coefficient analysis between MYOF and Rab11 in single cells, with different shaped points indicating individual cells from different replicates. In total 44 individual cells were analysed across 3 biological replicates. **C** siRNA-treated A549 Rab11-mCherry cells stained for MYOF 48 h post-transfection. Maximum intensity Z-stack projections are shown. Scale bar 10 μm. **D** Measurement of the largest MYOF signal accumulation in each cell, measured by automated thresholding based on image entropy. **E** Measurement of the largest Rab11-mCherry signal accumulation in each cell, measured by automated thresholding based on image entropy. For **D**, **E**, data were analysed using a two-tailed unpaired Student's $t$ test with *=$p < 0.05$ and ***=$p < 0.001$. All experiments consist of 3 biological replicates, with the bar representing the mean of the replicates and error bands representing standard deviation.

was also apparent in the nucleus, specifically during RSV infection, and this is the only time we observed nuclear localising MYOF during the course of our study. At least for RSV-infected cells, MYOF staining colocalised well with the Rab11-mCherry signal, confirming their relationship even during RSV infection.

These data confirm myoferlin as an active player in vRNP trafficking along the Rab11 recycling pathway for multiple enveloped viruses. As such, they provide novel insights into multi-species enveloped virus vRNA trafficking in respiratory cells and, by extension, intriguing targets for the design of broad-spectrum antivirals.

### MYOF likely recruits EHD1/2 for end-stage ICV biogenesis

Myoferlin is a large protein composed of several C2 domains, some of which were characterised as calcium binding, membrane binding, or protein binding. Specifically, the second C2 domain (C2B) was found to be responsible for direct interaction with Eps15 Homology Domain 2 (EHD2)[17]. To assess whether EHD2, or the closely related EHD1, are present within these ICVs, A549_Rab11-mCherry cells were transduced with lentiviral vectors expressing GFP-tagged forms of the protein of interest, or a negative control GFP-only (EV). They were then infected with wild-type WSN for 6 h, treated with nucleozin for 2 h, and imaged. In untreated cells, both EHD proteins localised to distinct cytosolic structures, including the perinuclear area where they partially colocalised with Rab11, and therefore likely myoferlin (Fig. 6A). In contrast, nucleozin treatment generated large and easily

recognisable Rab11 aggregates, with both EHD1 and EHD2 predominantly localising therein (Fig. 6A). These data confirm the presence of members of the EHD family of proteins, specifically EHD1 and EHD2, in ICVs. To further investigate the phenotypic effects of EHD2 expression in IAV infected cells, we turned to siRNA-mediated knockdown of EHD2 transcripts. First cell viability assays were performed with varying concentrations of EHD2-targeting siRNA, as performed previously for MYOF, with only the highest tested concentration of 10 nM demonstrating a significant reduction in cell viability (Fig. 6B). We proceeded with the lower concentration of 5 nM, transfected A549 cells for 48 h before subsequent infection with WSN at MOI3 for 8 h. On quantifying viral titres and RNA levels we observed a similar pattern to prior experiments, with significant reductions in titres in siEHD2 treated cells (Fig. 5C) but no corresponding changes in viral NP mRNA or vRNA levels (Fig. 5D). This indicates that EHD2, in a similar manner to MYOF and Rab11, play a critical role only in late stages of viral replication. Finally, due to previously published observations that knockout of MYOF expression negatively affects EHD2 levels in myoblasts[17], we investigated whether the same was true in A549s transfected with a MYOF-targeting siRNA. Indeed, we observed a clear reduction in EHD2 protein levels by Western blot in MYOF knockdown cells when compared to siNSC treated cells (Figs. 5E and S7). This led us to conclude that myoferlin likely disrupts the stability or proper membrane association of EHD2, a protein with which it functionally interacts in endocytic and membrane repair pathways. In the absence

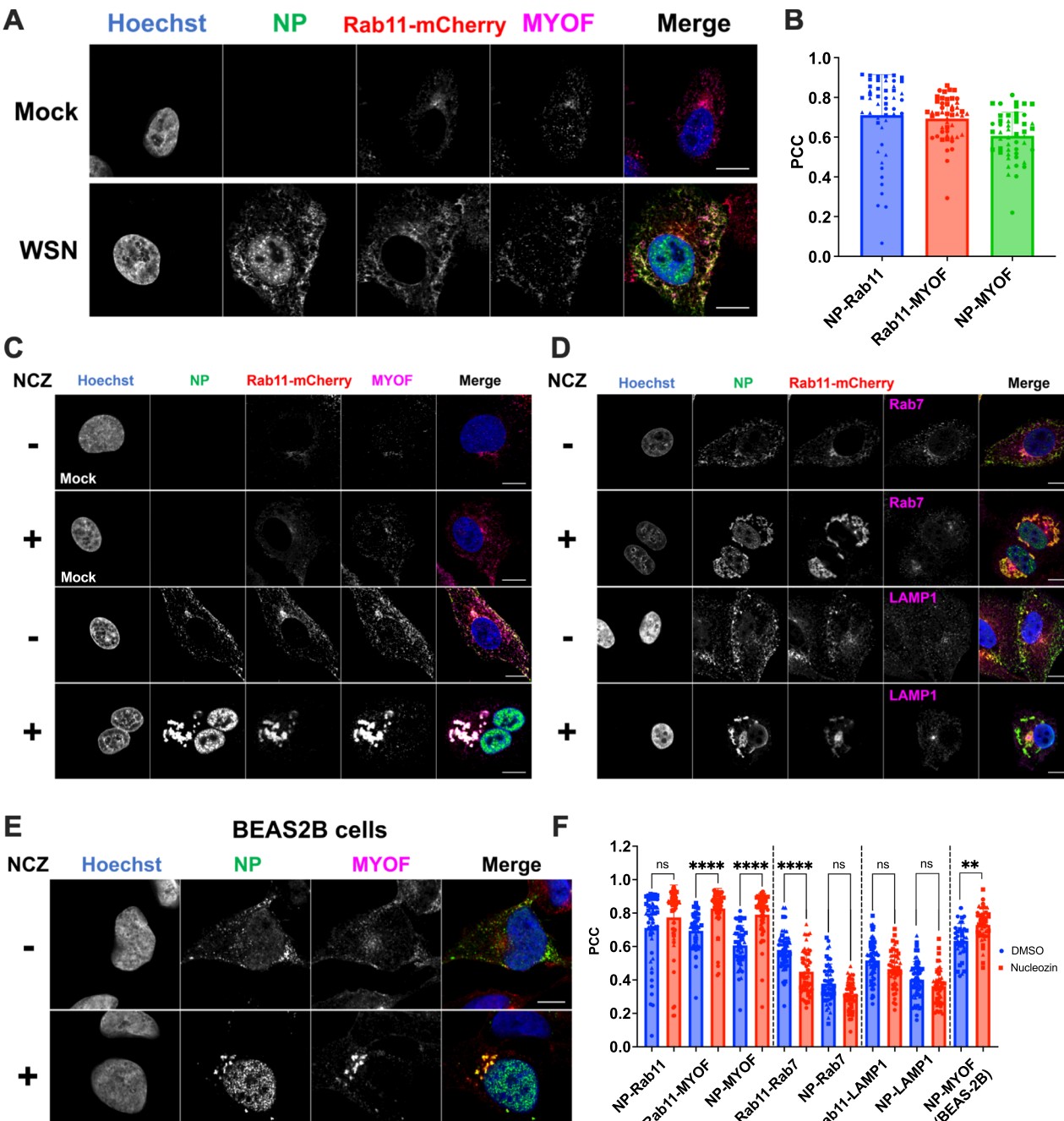

**Fig. 4 | MYOF is specifically retained in influenza A vRNP-trafficking vesicles.** **A** A549_Rab11-mCherry cells were mock-infected or infected with MOI 5 WSN for 8 h, then fixed and stained for NP (green) and MYOF (magenta). Scale bar 10 µm. **B** Pearson's correlation coefficient quantification of the pairwise colocalisation between NP, MYOF, and Rab11, in 49 single cells across 3 biological replicates. A549_Rab11-mCherry cells were infected with MOI 5 WSN for 6 h, treated or mock-treated with nucleozin (NCZ) for 2 h, then fixed and stained for NP and MYOF (**C**) or NP and either Rab7 or LAMP1 (**D**). Scale bar 10 µm. **E** BEAS-2B cells were infected with MOI 5 WSN for 6 h, treated or mock-treated with nucleozin (NCZ) for 2 h, then fixed and stained for NP and MYOF. Scale bar 10 µm. **F** Single-cell Pearson's correlation coefficient quantification of colocalisation between NP and MYOF, or NP and the controls (Rab7/LAMP1) in 49 single cells across 3 biological replicates. In blue are represented the mock-treated cells, and in red the nucleozin-treated. In **B**, **F**, different shaped points indicate individual cells from different replicates. For **F**, single-cell Pearson's correlation coefficients were analysed using a two-way ANOVA followed by a two-sided Tukey's multiple-comparisons test, with $p$ values adjusted to control the family-wise error rate at $\alpha = 0.05$ with **=$p < 0.01$ and ****=$p < 0.0001$. All experiments consist of 3 biological replicates, with the bar representing the mean of the replicates and error bands representing standard deviation.

of myoferlin, EHD2 may therefore become targeted for degradation, leading to the observed reduction in protein levels.

Given their known role in vesicular biogenesis and the association between EHD2 and MYOF[17], we propose that during infection with respiratory viruses, such as IAV, RSV, or SeV, EHD proteins are recruited to vRNP trafficking vesicles by MYOF to coordinate membrane remodelling and complete ICV maturation (Fig. 7). This would lead to trafficking of vRNPs to the cytoplasmic membrane for packaging into virions and subsequent egress.

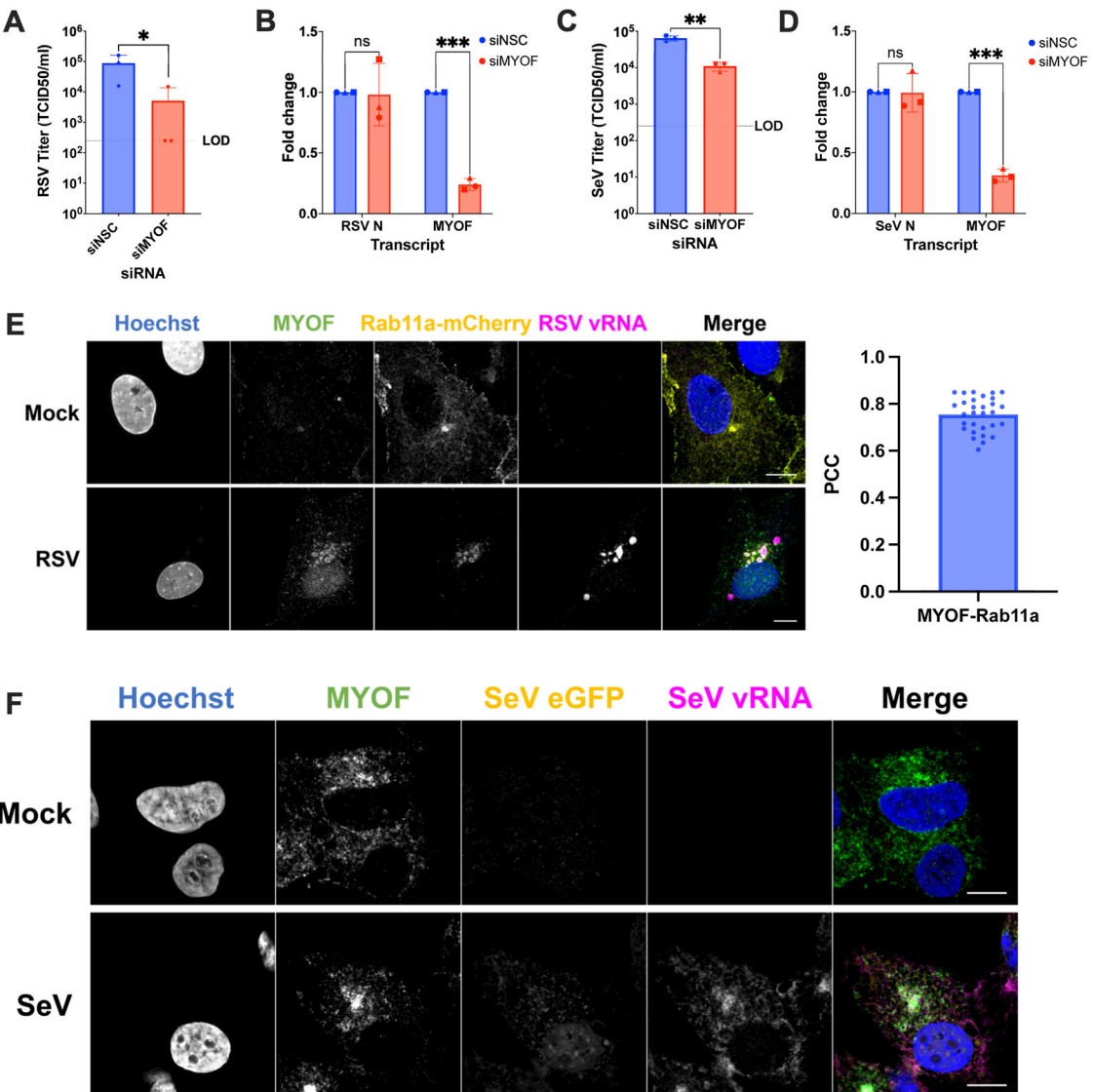

**Fig. 5 | Respiratory viruses commonly utilise Rab11-MYOF vesicles at late stages of infection. A, B** A549 cells were transfected with 4 nM of the indicated siRNA for 48 h, then infected with RSV_mKate2 at an MOI 2 for 24 h. Supernatants were then harvested and titres quantified by TCID50 (**A**) while RNA was extracted from cells and relative RSV N and MYOF mRNA levels determined by qPCR (**B**). **C, D** A549 cells were transfected with 4 nM of the indicated siRNA for 48 h, then infected with Sendai_eGFP at an MOI 2 for 24 h. Supernatants were then quantified by TCID50 (**A**), while RNA was extracted from cells and relative SeV N and MYOF mRNA levels determined by qPCR (**B**). **E** A549_Rab11-mCherry cells were mock-infected or infected with RSV_BT2a for 24 h, then processed for smiFISH to image the viral genomes (green), as well as immunofluorescence for MYOF (magenta). Scale bar 10 μm. The PCC between MYOF and Rab11 was then calculated to determine the strength of colocalisation across 30 individual infected cells across 3 biological replicates. **F** A549 cells were mock-infected or infected with Sendai_eGFP for 24 h, then processed for smiFISH to image the viral genomes (green), as well as immunofluorescence for MYOF (magenta). Scale bar 10 μm. For **A–D**, data were analysed using a two-tailed unpaired Student's $t$ test with *=$p < 0.05$, **=$p < 0.01$ and ***=$p < 0.001$. All experiments consist of 3 biological replicates, with the bar representing the mean of the replicates and error bands representing standard deviation.

## Discussion

Due to their relatively small genome sizes, and thus limited coding capacity, RNA viruses rely on host machinery to accomplish most, if not all, of the steps in their replication cycle. From the moment of infection, a series of intricate pathways are exploited and hijacked to ensure viral replication. IAV intracellular trafficking is very well-studied and an ideal model to investigate the dynamic movements of viral RNA genomes. Firstly, upon IAV virion binding to the plasma membrane sialic acid receptors, clathrin-mediated endocytosis is triggered (with a minority of particles entering through other means) and the virion is trafficked in the endosomal pathway[7]. Acidification of the compartment induces the release of vRNPs in the cytosol, with nuclear localisation signals on the vRNP recruiting importins to carry it across the nuclear pore[25]. After replication in the nucleus, newly formed vRNPs interact with M1 and NEP to hijack the CRM1 (XPO1) pathway for export to the cytosol[25,26]. Once through the nuclear pore complex, vRNPs rapidly concentrate close to the nuclear membrane in the vicinity of the microtubule organising centre (MTOC), and associate with Rab11+ vesicles through direct interaction of Rab11 with PB2[27,28]. Seminal publications from the Naffakh lab and the Amorim lab clearly established the extensive remodelling of the vesicular network upon IAV infection[7,16,29]. Electron microscopy and live-imaging experiments proved the formation of ICVs, characteristic vRNP-clad Rab11+ vesicles disrupting the normal ERC (Fig. 6B). ICVs are generated and trafficked along microtubules (MTs). Interestingly, the microtubule network does not seem to be strictly required for IAV replication, so there must be other factors at play[30]. ICVs then localise near the endoplasmic

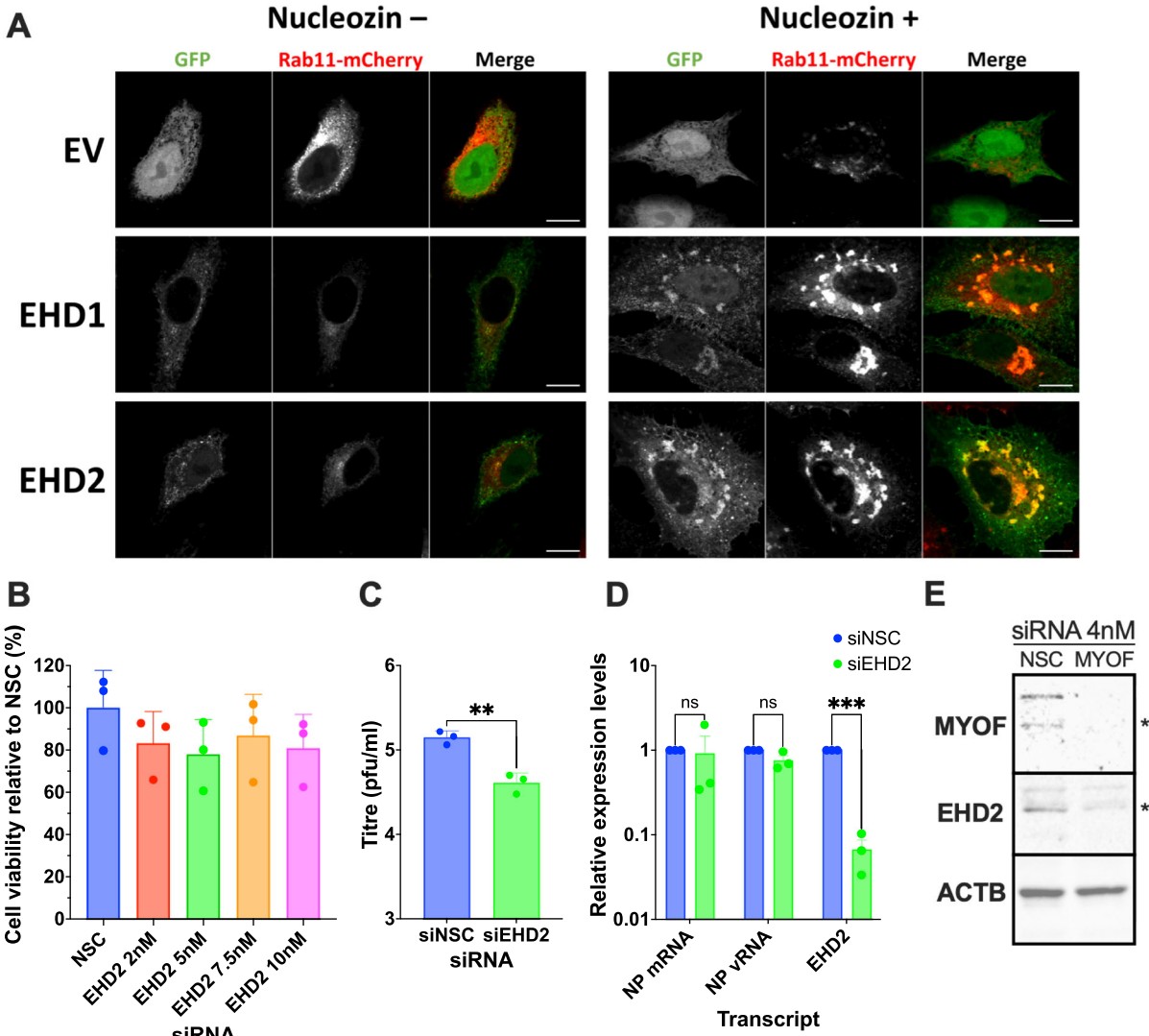

**Fig. 6 | MYOF likely recruits EHD1/2 to complete vRNP-trafficking vesicle biogenesis. A** A549_Rab11-mCherry cells were transduced with a vector expressing either GFP, EHD1-GFP or EHD2-GFP. They were then infected with MOI 5 WSN for 6 h and treated (left) or mock-treated (right) with nucleozin (NCZ) for 2 h before fixation and imaging. Scale bar 10 μm. Images are representative of 3 biological replicates. **B** A549 cells were transfected with the noted concentration of siEHD2 or 10 nM siNSC for 48 h before cell viability was determined by a resazurin assay, n = 3. **C, D** A549s transfected with 5 nM of the indicated siRNAs were then infected after 48 h with WSN MOI 3 for 8 h prior to titration of conditioned media (**C**) or RNA extraction and subsequent qPCR to quantify NP mRNA, NP vRNA and EHD2 RNA levels, n = 3. **E** Western blot of MYOF, EHD2 and ACTB protein expression in A549 cells 48 h post-transfected with 5 nM of either NSC or MYOF targeting siRNA. * indicates the appropriate protein band within each panel when multiple bands are visible. Uncropped images are shown in Fig. S9. For **C, D**, data were analysed using a two-tailed unpaired Student's t test with **=p < 0.01 and ***=p < 0.001. All experiments consist of 3 biological replicates, with the bar representing the mean of the replicates and error bands representing standard deviation.

reticulum (ER), at ER egress sites (ERES), where they are unloaded from MTs by ATG9A[7,15,16]. The high local concentration of vRNPs triggers liquid-liquid phase separation (LLPS), in which free vRNPs and ICVs coalesce in a membraneless, liquid-like state. These liquid vRNP droplets are believed to be sites for vRNP reassortment and possibly bundling[15]. Bundles of vRNPs are then transported to budding sites at the plasma membrane. The mechanisms surrounding this last trafficking step are still unclear, however, vRNPs are thought to bind M1 at the cytosolic face of specialised lipid rafts enriched in cholesterol and viral glycoproteins. Molecular crowding, as well as specific membrane-bending activities of M1, M2, HA and NA, induce budding and scission[31].

These processes completely overhaul the trafficking networks of the cell, with the involvement of the ERC, the ER, the lysosomal pathway, and the Golgi apparatus. Hence, an important focus of IAV

research is on characterising the exact composition of vRNP trafficking vesicles.

To this end, we proceeded to perform an interactome study to investigate the host factors that are essential to the late stages of trafficking for IAV vRNPs (Fig. 1A). Through IP-MS at an early (6 hpi; Fig. 1D) and late (16 hpi; Fig. 1E) timepoint, coupled with PFA cross-linking and RNase digestion, we generated a snapshot of proteins and pathways associating with the influenza polymerase complex (FluPol) (Fig. 1F). The FluPol is a dynamic entity that interfaces with a multitude of cellular processes over the course of a single infectious cycle, with our method capturing the polyhedric nature of this. By directly comparing early to late datasets we could then further demarcate the host factors likely contributing to vRNP trafficking, bundling, and packaging (Fig. 1G). Previous studies aimed to characterise IAV interactomes, mainly by expression of individual or subsets of viral proteins, or in

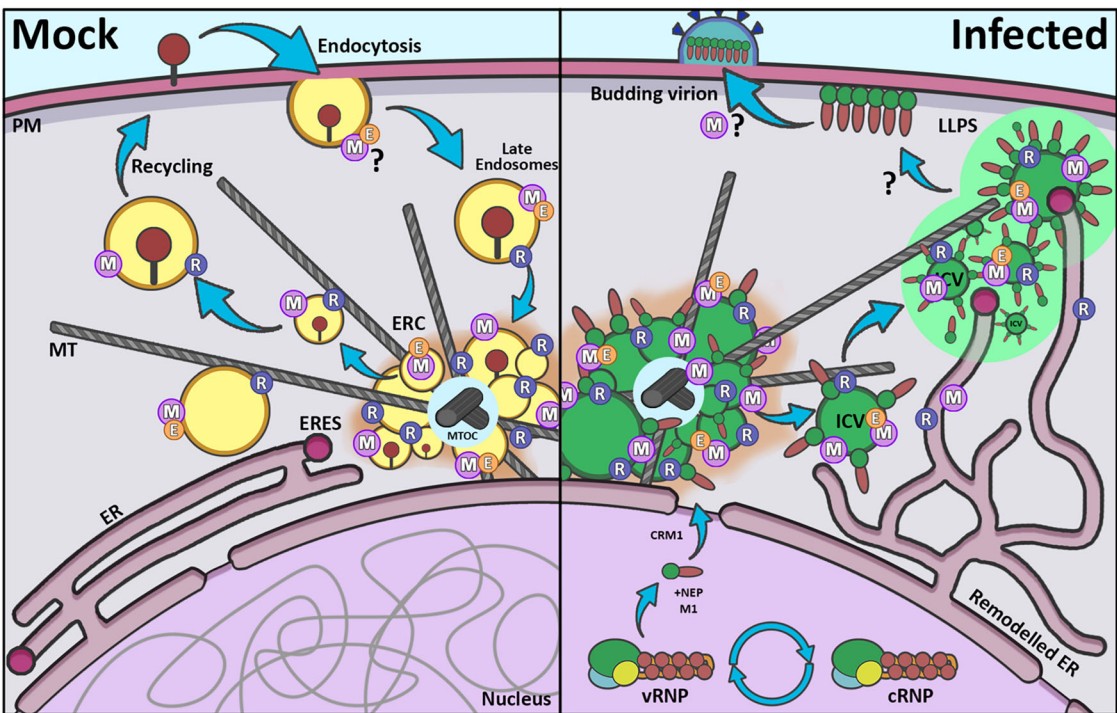

**Fig. 7 | Model of MYOF and EHD2's role in Rab11-mediated recycling and vRNP trafficking in infected cells.** Schematic showing MYOF and EHD1/2 incorporation into vRNP-trafficking ICVs. On the left, in uninfected cells MYOF associates with Rab11 recycling endosomes at the ERC and is actively involved in the transport of cellular components including receptors and transferrin for recycling. On the right, in IAV infected cells the ER is remodelled while vRNPs exported from the nucleus congregate at the MTOC. These vRNPs are trafficked to the cellular membrane for

packaging on ICVs containing Rab11 and MYOF, with vesicular maturation possibly triggered by the recruitment of EHD1 and EHD2. ICVs are proposed to traffic to the ERES where vRNPs may then undergo LLPS prior to packaging into virions. PM plasma membrane, MT microtubules, ERC endosomal recycling compartment, ER endoplasmic reticulum, ERES ER egress sites, MTOC MT organising centre, M myoferlin, R Rab11, E EHD1/2, LLPS liquid-liquid phase separation, ICV irregularly coated vesicles.

infected cells without any crosslinking[32–36]. Our study forgoes the purification of exclusively direct interactors in favour of isolating more indirect, long-range interactions still fundamental for viral replication, in the context of the remodelled infected cell.

The quantification of proteins enriched in the late sample compared to controls, identified groups of proteins associated with intracellular protein transport, translation and protein folding (Fig. S1). This is likely due to the active translation and trafficking of mature FLAG-tagged PA, and the nuclear export of vRNP complexes. However, because this method of quantification can only quantify proteins present in both FLAG (WSN_FLAG) and control (WSN_WT) datasets, we were more interested in investigating the proteins enriched in the late compared to early post-infection timepoints. Here we found two clear groups of proteins: those involved in nucleocytoplasmic transport, as expected for exporting vRNP complexes and, most interestingly, 14 proteins previously described to be incorporated into WSN virions and released from the cell[20]. This validates our timepoint, clearly indicating the capture of late interactors of the FluPol complex, while substantiating the discovery of distinct packaged host factors into IAV virions. Interestingly, a few hits were strongly enriched in the late over early interactome and had no previously known roles in IAV replication. Our siRNA knockdown and infection assays, coupled with prior literature on its known functions, identified myoferlin as the main host factor of interest. Particularly, MYOF-depleted cells could support viral gene expression and genome replication but exhibited a reduction in infectious virus production, indicating a role during a post-replication step (Fig. 2B–D).

Myoferlin is a large multidomain protein, part of the ancient family of Ferlin proteins. These transmembrane factors are involved in vesicular trafficking and membrane dynamics in a myriad of contexts[22,37]. Myoferlin was first characterised as an important factor in

muscle cell maturation[38,39] but has since been linked to a plethora of processes in vastly different cellular environments. Studies have proven myoferlin's involvement in mitochondrial fission/fusion events[40], lysosomal membrane damage repair[41], endothelial growth factor receptor function[42], excitation/contraction coupling in mouse skeletal muscle[43], invasive phenotype in various cancer types[44], and endocytosis[22,45]. In fact, a recent study found that myoferlin ensures proper localisation of the Env protein to sites of virion budding in human T-cell leukaemia virus type 1 (HTLV-1) infection[46].

In this work, we show that myoferlin is an important component of the Rab11+ recycling endosomes, colocalising extensively with the GTPase in uninfected lung epithelial cells, influencing its localisation and its function, while MYOF's localisation is itself affected by Rab11.

Endosomal recycling is a fundamental host pathway involved in the vesicular trafficking of a myriad of cargo, such as internalised receptors and other endocytosed material[14]. Under normal growth conditions recycling endosomes spread out in the cytoplasm but are concentrated in a characteristic juxtanuclear accumulation, with myoferlin localisation closely resembling this typical recycling endosome pattern (Fig. 3A). Additionally, it was reported previously that myoferlin-null myoblasts display delays in transferrin turnover in pulse-chase experiments, which we also observed by IF in A549 knockdown cells (Fig. S4)[17]. Collectively, these data led us to hypothesise that myoferlin is an important component of endocytic recycling in lung epithelial cells, which is usually the target for infection by respiratory viruses such as IAV and RSV. Using A549 cells as a model we discovered that Rab11 and myoferlin are intimately connected through their roles in recycling, with each dependent on the other for normal cellular distribution (Fig. 3C–E). With the interdependence of these two components established, coupled with the knowledge that Rab11-mediated trafficking of vRNPs is a critical step in IAV replication, it is

clear that the inhibitory effect of myoferlin loss on viral titres alongside no effects in viral RNA levels may be attributed to this late-stage in the viral replication cycle.

Mounting evidence suggests that several RNA viruses exploit similar strategies to accomplish directional intracellular trafficking of their vRNPs[4,6,8,47]. Given the strong link that we established between Rab11 and MYOF both in uninfected cells and during IAV infection, we reasoned that it was likely that other pathogens that hijack this pathway would also require myoferlin for efficient replication. Indeed, myoferlin knockdown followed by RSV or SeV infection greatly reduced overall infectious virus release (Fig. 5A, C) while viral RNA levels remained unaffected (Fig. 5B, D), mirroring the results observed following IAV infection (Fig. 2C, D). Furthermore, upon imaging of RSV vRNA, Rab11-mCherry and MYOF, a clear localisation of all three signals to multiple defined cytosolic spots was evident (Fig. 5E). Again, this is a phenotype recapitulated in SeV-infected cells (Fig. 5F). Interestingly, it appears as though the largest cytosolic spots, likely active viral replication factories, do not colocalise with Rab11 or myoferlin. This implies that both Rab11 and myoferlin are not required for actual transcription or replication, as we would expect, but instead are only recruited to or associate with actively trafficking mature vRNP-containing vesicles. This clearly aligns with our observations and those of other studies investigating the role of Rab11 strictly in vRNP egress. While only three respiratory viruses that utilise Rab11-mediated vRNP trafficking were investigated here, these data imply a conserved role for myoferlin in the replication of other Rab11-dependent viruses, including respiratory pathogens of the Coronaviridae and Paramyxoviridae families[3,10,48].

In establishing myoferlin as an important component of recycling in healthy cells and vRNP trafficking in a range of viral infections, we were interested to note that the second C2 domain (C2B) was previously reported to directly interact with EHD2[17]. The EH domain is a protein-protein interaction domain often associated with vesicular trafficking factors. It functions by binding cognate NPF/KPF motifs in partner proteins. EHD proteins themselves contain both the eponymous EH domain and the NPF/KPF motif, allowing for self-oligomerisation[49]. The formation of multimeric EHD structures drives membrane remodelling events and contributes to vesicular trafficking. Myoferlin itself contains an NPF/KPF motif in its second C2 domain, which was found to be fundamental for EHD2 interaction[17]. Therefore, we hypothesised that, with the presence of MYOF in the ERC, but more importantly within ICVs during IAV infection, members of the EHD family of proteins would be recruited to these vesicles to trigger membrane remodelling and end-stage vesicular maturation (Fig. 7). Indeed, as shown in Rab11-mCherry cells expressing eGFP-tagged EHD1 and EHD2, upon IAV infection both proteins were found to be localised close to Rab11 at the MTOC. This observation was even starker upon nucleozin treatment to aggregate vRNPs and their accompanying interactors (Fig. 6A). Moreover, EHD2 depletion was accompanied by a reduction in infectious virus production, uncoupled from genome replication (Fig. 6C, D). Additionally, in alignment with a previous study, we observed that a reduction in MYOF protein levels negatively impacted EHD2 protein levels (Fig. 6E). These findings suggest that myoferlin plays a critical role in maintaining EHD2 stability and proper membrane localisation, likely through their functional interaction in endocytic and membrane repair pathways, such that, in its absence, EHD2 becomes destabilised, resulting in decreased protein levels. Overall, our investigation into EHD-MYOF dynamics has led us to propose that the membrane remodelling observed, particularly during IAV infection, may be triggered by EHD proteins after their recruitment to the ERC by myoferlin. Interestingly, it has also been speculated that the membrane remodelling itself could be the driving force behind vRNP movement towards the cell membrane[7]. However, very little work was carried out to explore the role of EHDs in viral infections, with only limited evidence supporting EHD1 involvement in Vesicular Stomatitis Virus Glycoprotein (VSV-G) localisation to the plasma membrane[50].

As stated, important research is being carried out to fully characterise the makeup of influenza trafficking vesicles. So far very little progress has been made towards this goal, but this manuscript extends the list by three, adding MYOF, EHD1 and EHD2. All these factors clearly colocalise with ICVs during trafficking and aggregate upon nucleozin treatment. More work is needed to completely elucidate their role in infection, as well as their involvement in the replication cycle of other pathogens.

Interestingly, MYOF ablation also seemed to affect viral mRNA and vRNA production (Fig. 2D), in contrast to the established literature around Rab11 and IAV. As discussed above, myoferlin is believed to play a role in endocytosis as well as recycling, and we believe that its importance in the viral replication cycle is not limited to late trafficking. Myoferlin in silico interactome mapping (Fig. S5) revealed the extent of MYOF's involvement in membrane processes, interacting with endocytosis, recycling, membrane repair and trafficking factors. Interestingly, one of the main hits of this STRING interactome mapping is Annexin-A1 (ANXA1), which was also among the enriched hits in our late PA interactome. ANXA1 was found to be an important determinant of vRNP trafficking from endosomes to the nucleus[51], again connecting MYOF, endocytosis and vRNP trafficking. Future work aims to better characterise MYOF during the earlier steps of infection.

At the other end of the infection cycle, myoferlin was shown to localise to plasma membranes and specifically lipid rafts, where it interacts with Dynein-2 to ensure membrane resealing after fission-based endocytosis[22]. As stated above, specialised lipid rafts are the sites on the cell surface at which IAV budding occurs[52]. An attractive hypothesis puts myoferlin at the crossroads of membrane remodelling events throughout the viral replication cycle, from endocytosis to ICV remodelling to budding at the plasma membrane.

Novel, broadly applicable therapeutic interventions provide the greatest cost-to-benefit ratio for biomedical research. One avenue in the design of such a strategy is the discovery and subsequent targeting of a ubiquitous host protein essential during a key stage in the replication cycle of various clinically relevant human viruses. One such seemingly pervasive stage is the egress of viral genomic RNA exploiting the endosomal recycling pathway. Herein, we describe the intimate relationship between Rab11-containing vesicles and myoferlin, under normal growth conditions and within IAV-induced remodelled ICVs. Importantly, however, this relationship is conserved on RSV and SeV vRNA-trafficking vesicles. While the data discussed here directly present strong evidence for the importance of myoferlin for vRNA trafficking of IAV, RSV and SeV specifically, they imply that this would also be the case for all Rab11-dependent viruses, including coronaviruses[10], parainfluenza viruses[9], Hantaviruses[4], ebolaviruses[6] and flaviviruses[5].

## Methods

### Cells
For this work immortalised A549 (ATCC; CCL-185), HEK 293 T (ATCC; CRL-3216), BEAS-2B (ATCC; CRL-3588), Hep-2[53] and MDCK (CCL-34) cells were used. All cells were cultured in DMEM supplemented with 1% Pen-Strep (Thermo Fisher Scientific; 15140122) and 5% FBS (Thermo Fisher Scientific; 10270106) at 37 °C and 5% $CO_2$.

### IAV virus stocks and infections
WSN (A/WSN/33) viral stocks were generated from a reverse genetics system that has been described previously[54]. WSN_PA-mNeon virus was generated from the same reverse genetics system, where the pPolI_PA plasmid was substituted for the pPolI_PA-mNeon plasmid encoding for a C-terminal mNeon followed by a duplicated packaging signal, and has been described previously[55]. Additionally, a C-terminally FLAG-tagged PA virus was also rescued from the same system, pPolI_PA-FLAG. Cal09 (A/California/7/2009), Shanghai90 (A/Shanghai/24/

1990), Brisbane18 (A/Brisbane/02/2018) viruses were grown from isolates acquired from the National Institute for Biological Standards and Control, UK. All stocks were grown on MDCK cells in IAV growth media consisting of DMEM, Pen-Strep, 0.2% BSA (Merck; A8412), 25 mM HEPES (Merck; H0887) and TPCK-trypsin (Merck; T8802), and titrated on MDCK cells by plaque assay.

## Immunoprecipitation for LC-MS/MS

For the late timepoint influenza interactome experiment, approximately $2*10^7$ ($4*10^7$ for the early interactome) A549 cells per condition were seeded on 15 cm plates 24 h prior to infection. The next day, cells were infected with IAV WSN_PA-FLAG (FLAG) or A/WSN/33 (WT) at an MOI of 1, then fixed at 16hpi (6hpi for the early interactome) by a 10' incubation with 1% PFA in PBS. Residual PFA was quenched with a 10' treatment with 1.25 M Glycine in PBS and washed three times with cold PBS. Cells were then scraped and pelleted. The samples were lysed in RIPA buffer supplemented with protease inhibitors (Merck; 4693116001) for 1 h on ice. Afterwards, cell debris was removed by centrifugation and 10% of the remaining sample was taken as input.

The immunoprecipitation was carried out using magnetic protein G beads (Thermo Fisher Scientific; 88848), which were washed and incubated with monoclonal anti-FLAG antibodies (Sigma; F1804) for 45' before use. We employed 100 μl of beads, 50 μl of antibodies, and 10 μl of RNAse Cocktail (Thermo Fisher Scientific; AM2286) per condition. After capturing for at least 2 h, the beads were thoroughly washed in buffers of decreasing stringency: RIPA buffer, high salt buffer (5% HEPES-KOH, 50% KCl, 0.05% NP40 in $H_2O$), and PBS; all ice-cold and supplemented with 1% DTT. After removing 10% of the beads for western blotting, the rest was resuspended in $H_2O$ and sent for mass spectrometry analysis. Three independent biological replicates were produced and analysed for each condition.

## Western blot

Western blotting was performed as previously described[56]. The antibodies used can be found in Supplementary Table 1. Protein bands on blots were visualised by chemiluminescence.

## Silver staining

Silver staining was performed according to the manufacturer's protocol, using the Pierce Silver Stain Kit (Thermo Fisher Scientific; 24612), using samples processed as detailed in the IP-MS paragraph.

## LC-MS/MS and data analysis

Mass spectrometry was performed at the Cambridge Centre for Proteomics. Sample treatment, preparation and trypsin digestion were performed exactly as described elsewhere[55]. Peptide pools were quantified by a Qubit protein assay (Thermo Fisher Scientific). Data were acquired on an Orbitrap™ Fusion™ Lumos™ Tribrid™ mass spectrometer with an equal amount of each sample sequentially loaded over individual 60-min runs with no fractionation given the likely low complexity of the sample. Raw mass spectrometry data were analysed using MaxQuant software (v2.4.7.0) and utilising the Andromeda search engine[57]. The early (6hpi) and late (16hpi) interactomes were captured separately so pairwise comparisons were analysed individually. The MS/MS spectra were aligned to the Uniprot Homo sapiens database and the corresponding influenza virus Uniprot protein database. Known mass spectrometry contaminants and reversed sequences were also included. The search was performed with trypsin selected as the specific enzyme and a maximum of 2 missed cleavage events allowed. The MS accuracy was set to 10 ppm, then 0.05 Da for the MS/MS. The maximum peptide charge was set to seven and seven amino acids were required for the minimum peptide length. One unique peptide to the protein group was required to call the presence of a specific protein, while a false-discovery rate (FDR) of 1% was set. Statistical analysis to ascertain which proteins are enriched in one set

of biological samples over another was performed using the intensities quantified by MaxQuant on Perseus software (v2.0.11)[58]. First, any potential contaminants, reverse hits or proteins that were not well identified (above 1% FDR) were removed. After filtering, the intensities were log-transformed (log2) and entries absent from at least 70% of the replicates for a given condition were removed. Missing values were replaced from the normal distribution. The data were then normalised by subtracting the most frequent value and enrichments between conditions were calculated. Statistical significance was calculated by way of an unpaired two-tailed Student's $t$ test for each protein group. These values were then log-transformed (−log10) and the resulting $p$ values and enrichments were graphed as volcano plots. The cutoff for statistical significance was set to $p < 0.05$. Analysed data is included in Supplementary Data 1–4.

## siRNA knockdowns

For all siRNA knockdowns, siRNAs (TriFECTa DsiRNA Kit, IDT) targeting the specified mRNA, or a non-specific control siRNA were used. These siRNA sequences are listed in Supplementary Table 2. A final concentration between 20 nM and 2 nM total siRNA was used in each condition according to the experiment, and siRNAs were all transfected using Lipofectamine RNAiMAX (Invitrogen; 13778075) following the manufacturer's instructions. A549 cells were seeded at the time of transfection and incubated for 24 h before the media was changed to fresh growth media. Cells were then incubated for an additional 24 h before further processing.

## Growth curve

A549 cells were cultured for 24 h prior to infection at MOI 0.01 with either wild-type A/33/WSN or WSN_PA-FLAG. After 1 h inoculation at 37 °C, cells were washed in PBS and supplemented with IAV growth media. The input sample consists in the supernatant of infected cells harvested immediately at this step. The set of supernatants was titrated on MDCK cells as described.

## IAV qPCR analysis

A549 cells previously transfected with siRNAs for 48 h, were infected with A/WSN/33 virus at an MOI 3 PFU/cell. At 8 hpi RNA was extracted from cells using TRIzol (Invitrogen; 15596026). RNA was isolated and precipitated following the manufacturer's instructions and 200 ng was reverse transcribed into cDNA using the ABI cDNA synthesis kit (Applied Biosystems; 4368814). For cellular targets GAPDH and MYOF, RT was performed using an oligo dT, while for viral NP mRNA and NP vRNA specific primers were used[59]. These primers are listed in Supplementary Table 3 alongside the primers used for qPCR amplification. All qPCR experiments were performed using SYBR Select Master Mix (Applied Biosystems; 4472908) following the manufacturer's instructions. All qPCR data were normalised to GAPDH and quantified using the ΔΔCT method in relation to the siRNA NSC transfected cells.

## Immunofluorescence

Approximately $1.5*10^5$ A549 (or A549_Rab11-mCherry) cells were seeded on glass coverslips (thickness #1.5) for 24 h prior to treatment. For experiments involving knockdown, siRNAs were transfected as stated above during seeding and cells were cultured for 48 h before further treatment.

Cells were then infected/treated according to the experiment, then fixed in either PFA 4% for 10' followed by permeabilization by Triton X-100 0.2% for 20', or ice-cold methanol for 5'. The latter proved much better for the visualisation of MYOF and was used predominantly in experiments with wild-type A549s. Samples were then blocked in 3% BSA for 30', followed by overnight incubation at 4 °C with the primary antibodies diluted in 3% BSA. The next day, cells were washed and incubated for 1 h with the secondary antibodies diluted in

3% BSA, then counterstained with Hoechst (Abcam; ab228551, 1:4000) before being mounted on slides using ProLong Diamond Antifade Mount (Thermo Fisher Scientific; P36961). Antibodies and dilutions are listed in Supplementary Table 1.

### smiFISH

For smiFISH experiments, we employed the protocol described in Tsanov et al., using primary probes we designed (Supplementary Table 4, IDT oligo pool), annealed to the published imager strands (IDT Cy3- or Cy5-conjugated oligos)[60]. After probe hybridisation and subsequent washes, we proceeded with blocking and IF as described above.

### Lentiviral overexpression cell line generation

The Rab11-mCherry, EHD1-eGFP and EHD2-eGFP cell lines were generated by transduction with third-generation lentiviral vectors. Briefly, the producer 293 T cell line was transfected with a 3:2:1 ratio of d8.74:pMD2.G:lentiviral vector, using PEI according to the manufacturer's protocol. The lentiviral vector expressing mCherry-tagged Rab11 was pHR-FKBP:mCherry-Rab11 (Addgene plasmid # 72902; http://n2t.net/addgene:72902; RRID:Addgene_72902). The lentiviral vectors expressing eGFP-tagged EHD1 and EHD2 were cloned into the lentiviral vector pLCE and generated by PCR using a synthetic gene block as the template for EHD1 and the EHD2-mEGFP plasmid (Addgene plasmid #45932; http://n2t.net/addgene:45932; RRID:Addgene_45932) as the template for EHD2. The day after, the transfection media was exchanged for fresh media and incubated at 37 °C for 72 h. Then, the infectious supernatant was filtered with a 0.45 μm filter and overlaid onto fresh A549 cells. After 72 h, Rab11-mCherry cells were subjected to clonal isolation by dilution, cultured and screened for localisation of fluorescence signal, while EHD1-eGFP and EHD2-eGFP cell lines remained polyclonal for infection experiments.

### Transferrin uptake assay

For the transferrin uptake experiments, approximately $7.5*10^4$ A549 cells per condition were seeded and transfected with the appropriate siRNA as described above. A total of 48 h post-transfection, the cells were serum starved (in serum-free media) for 30' at 4 °C, then incubated with fluorescently labelled transferrin at a concentration of 7.8 μg/ml for 30' at 37 °C. Cells were then washed with ice-cold PBS and fixed with 4% PFA for 10', before further processing for IF.

Human holo-transferrin (Merck; T0665) was conjugated to Atto 647 N NHS ester (Merck; 18373), in sodium carbonate buffer, according to manufacturer's instructions. Excess dye was removed using Zeba™ Spin Desalting Columns (Thermo Fisher Scientific; 89877).

### Image analysis

Coverslips were imaged with a Leica Stellaris SP8 confocal microscope with a 100x objective with 1.4 NA. Detectors were set according to the dyes used, and each experiment had bleed-through controls to ensure signal specificity.

### All analysis was carried out in FIJI using custom macros

Briefly, for colocalisation analysis, the BIOP implementation of the JACoP plugin was used to quantify single-cell level pairwise correlation between signals in single z-slices after nuclear masking[61]. At least ten cells/condition per replicate were analysed across three biological replicates.

For the relocalisation experiments and the transferrin uptake assay, whole z-stacks were analysed by maximum projection, followed by automatic entropy-based thresholding of the signal of interest, and measurement of the area of signal foci. The resulting datasets were filtered for outliers in GraphPad Prism using the ROUT method and $Q = 1\%$. Roughly 30 cells/condition per replicate were analysed across three biological replicates.

### Nucleozin treatment

Nucleozin (MedChem Express; HY-50001) was reconstituted and stored in DMSO at a concentration of 5 mM. Mock/infected cells were treated with 1 μM nucleozin in infection media for 2 h prior to fixation.

### Cell viability assay

A549 cells were transfected with the appropriate siRNAs as stated for 48 h. The wells were then washed and treated with resazurin (0.15 mg/ml, Merck) for 1 h at 37 °C. Afterwards, part of the supernatant was transferred to a black 96-well plate for fluorescence reading. The parameters used were excitation 560 nm, emission 590 nm. The analysis was carried out on a FLUOstar Omega plate reader.

### Treatment with WJ460

A549 cells were first infected, as described above, at an MOI 3 with WSN for 2 h. Cells then underwent treatment with either DMSO or 100 nM WJ460 in infection media, described above. At 8 hpi supernatants were then harvested and virus titrated by plaque assay, while RNA was extracted from cells using TRIzol prior to cDNA conversion and quantification by qPCR.

### RSV/SeV infection and titre

Isolation and characterisation of the clinical isolate RSV-BT2a was previously described[62]. RSV A2/mKate2 was rescued by the Power Group from an infectious clone and helper plasmids kindly provided by Dr Martin Moore (Emroy University, Atlanta, USA).

### Sendai_eGFP was rescued as previously described[63]

For the titration experiments, approximately $1.5*10^5$ A549 cells per condition were seeded and transfected with the appropriate siRNA as stated above. A total of 48 h post-transfection, cells were infected with MOI 2 RSV_mKate2, or MOI 2 SeV_eGFP, for 24 h. Supernatants were then collected and titrated. Hep-2 cells were used for RSV titrations, as previously described[53].

For imaging, A549_Rab11-mCherry cells were infected with MOI 2 RSV wild type BT2a or SeV_eGFP for 24 h and processed for smiFISH and IF as described above.

### Statistical analysis

Unless otherwise stated, all experiments were performed as three independent biological replicates. When bar graphs are used to represent the data, each individual replicate can be observed as a point overlaid on the bar. Significance was calculated by performing two-way ANOVA with * indicating $p < 0.05$, ** indicating $p < 0.01$, *** indicating $p < 0.001$, and **** indicating $p < 0.0001$.

### Reporting summary

Further information on research design is available in the Nature Portfolio Reporting Summary linked to this article.

## Data availability

LC-MS/MS raw data are available on the PRIDE repository under the accession number PXD052709, while processed data are available in the Supplementary Information. The microscopy data generated in this study have been deposited in the Zenodo database under https://doi.org/10.5281/zenodo.18231359, while raw data used to generate graphs within this manuscript are available in the Source Data File. Source data are provided with this paper.

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

## Acknowledgements

This research was funded in part by an ERC-STG grant, PTFLU 949506 awarded to D.G.C. This research received infrastructure support from the Wellcome-Wolfson Institute for Experimental Medicine at Queen's University Belfast. We would like to thank Adam McShane and Dr Dessi Malinova for providing the Atto 647N-conjugated transferrin used in the transferrin uptake experiments.

## Author contributions

Conceptualisation: S.B. and D.G.C.; Methodology: S.B., H.L.T., S.S., E.P., H.L.C., J.J.M., C.H., E.M.P.E.G., O.T., and J.B.; Formal analysis: S.B., H.L.T., S.S., E.P., H.L.C., J.J.M., C.H., E.M.P.E.G., O.T., and J.B.; Investigation: S.B., H.L.T., S.S., E.P., H.L.C., C.H., E.M.P.E.G., and O.T.; Writing—original draft: S.B. and D.G.C.; Writing—review & editing: S.B., H.L.T., S.S., E.P., H.L.C., J.J.M., C.H., E.M.P.E.G., O.T., J.B., U.F.P., and D.G.C.; Supervision: U.F.P. and D.G.C.; Funding acquisition: D.G.C.

## Competing interests

The authors declare no competing interests.
