## [Transparent Peer Review file · Nature Communications]

Myoferlin is a component of late-stage vRNP trafficking vesicles for enveloped RNA viruses.

Corresponding Author: Dr David Courtney

Version 0:

Reviewer comments:

Reviewer #1

(Remarks to the Author)

The endosomal recycling pathway regulated by Rab11a is known to be responsible for the trafficking of vRNPs to the plasma membrane assembly sites of many RNA viruses. In this study, the authors performed IP-mass spec analysis to identify proteins associated with influenza A virus vRNP using WSN strain expressing Flag tagged PA. It is important to reveal the precise mechanism of how respiratory RNA viruses translocate vRNPs to the assembly site. However, presented data of this manuscript are very limited. Basically, their data show that MYOF is associated with Rab11a, which is known to interact with IAV vRNP. Unfortunately, there is no convincing data showing that MYOF is required for vRNP trafficking of IAV, RSV or SeV. The key conclusion is not supported by solid evidence as explained below.

- Fig 1E: Rab11a is well recognized to interact with vRNP. However, it seems that Rab11a was not detected in this IP-mass spec analysis (Supplemental Table S2). The authors need to explain why they failed to detect Rab11a in their analysis.
- Fig 1: Because this study aims to determine the interacting partner of PA, it will be informative if the authors provide data on the nuclear/cytoplasmic distribution of PA-Flag at these timepoints.
- Fig 2A and B: These data of low MOI (0.01) infection are not informative for the purpose of this experiment. To determine whether these proteins are required for virus assembly, they should infect at high MOI, such as MOI=5 as in Fig 2C, so that they can directly address the effect on a single step virus growth.
- Fig 2C: There is no significant reduction of virus assembly and release in cells transfected with MYOF siRNA at 2nM. This is a serious concern because at this concentration, MYOF was undetectable in cells as shown in Fig 3C. If this is true, MYOF is not required for the assembly and release of IAV. The same results were obtained with RSV and SeV in Figure 5B and D.
- Fig 2C and D. These results indicate that reduced virus production from cells transfected with MYOF siRNA at 10 nM is due to suppressed viral replication, not impaired vRNP trafficking.
- Fig 3D and E. The IF images of Rab11-mCherry and MYOF should be included in Fig 3D and E, respectively.
- Fig 5: The authors need to determine the effect of siRNA transfection on viral protein synthesis and genome replication. The reduced titer of progeny virions could be due to limited viral replication and protein production, not due to impaired vRNP trafficking.

Reviewer #2

(Remarks to the Author)

OVERVIEW:

Influenza A virus is a significant human pathogen. This study investigates the assembly of the virus, focusing on the packaging of its segmented genome (8 different vRNPs that form a complex), a process still not fully understood. There are two distinct steps critical for genome packaging. One is the formation of the genomic complex, and the other is the transport of each individual vRNP (or of the genomic complex) to sites of virion assembly at the plasma membrane. The order and these steps are difficult to disentangle experimentally and are not established yet. They have been linked to events that remodel the endoplasmic reticulum (ER), a process that leads to the formation of vRNP containing irregularly coated vesicles (ICVs) and also to an impairment in vesicular transport, as shown by the transferrin pathway. ICVs and other vesicles positive for Rab11a interact with vRNPs at ER exit sites (ERES) in viral inclusions, forming a liquid environment that may aid genome assembly. Identifying other cellular factors in these viral inclusions is essential for understanding the

molecular underpinnings of influenza genome assembly and transport to the plasma membrane, which is crucial for the formation of epidemic and pandemic infectious particles.

The paper reports that Myoferlin (MYOF) binds to vRNPs and localizes to Rab11a-positive vesicles and/or ICVs during IAV infection. This finding on its own is significant, but experimentally it is also well-supported by the data. MYOF and other vRNP-binding factors were identified through a robust pull-down followed by mass spectrometry (MS) analysis. Knockdown of these factors negatively impacted influenza replication, as evaluated by plaque assays. The association of MYOF with Rab11-positive vesicles and vRNPs was confirmed via immunofluorescence in both mock-infected and infected cells. Enhanced colocalization among vRNPs, MYOF, and Rab11 was observed with the addition of nucleoizin, an isoxazolylopyperazine compound known to aggregate NP. This section of the paper is strong, identifying a key component in ICVs.

However, the authors' attempt to elucidate how MYOF functions are less robust and require improvement, especially to disentangle MYOF's potential role in viral entry and assembly. Indeed, an effect at the early stages of infection could fully account for the observed reduction in viral titers (see comments below). It is possible that MYOF localizes with Rab11a and vRNPs but is hindered in performing its specific function. To address this concern, the authors present some evidence indicating that downstream factors EHD1/2 are recruited to Rab11 and vRNP sites. However, these data need to be more robust, and the functional relevance of this recruitment has yet to be demonstrated. The paper suggests that MYOF is an essential co-factor for Rab11-mediated recycling in IAV-infected cells, aiding vRNP trafficking—a pertinent question. The authors extend their findings to Sendai virus and Respiratory Syncytial Virus, which is commendable.

MAJOR ISSUES:

1) Given that MYOF is involved in many cellular functions, including endocytosis, the authors opted to elucidate the role of MYOF in uninfected cells. This has some shortcomings that should be resolved. In uninfected cells, when MYOF is depleted, Rab11a accumulates as in infection, so hence, at later stages, MYOF may actually not be needed for infection. One way to go around the involvement of MYOF in steps other than entry is to validate findings using drugs that blocks MYOF and add it at late stages of infection, thus evaluating the role of MYOF at this step. There are at least 2 drugs available that target MYOF – YQ456 and WJ460. Viral titres, localization of Rab11a and vRNPs by IF would be critical. Then, it would be important to evaluate if vRNPs are impeded from reaching the plasma membrane, which could be seen, for example, by transmission electron microscopy and RT-qPCR/HA-titres of released virions.

2) Influenza transcription and replication are impaired when MYOF is absent. Is MYOF important for influenza entry? Being important for viral entry, it is difficult to pinpoint whether the effects observed in viral titres relate to entry, assembly or both.

3) The authors looked at transferrin by microscopy-based approaches. This method is not quantitative. They found that transferrin gets arrested in the MTOC when MYOF is missing. However, as MYOF is also involved in endocytosis, it is unclear whether this result relates to a delay in the entry of transferrin. This assay should be done using FACS and a time course of transferrin recycling – 30 min, 1 and 2 hours and quantifying exactly how much transferrin is endocytosed and how much reaches the plasma membrane when MYOF is absent.

4) EHD1/EHD2 are known interactors of MYOF and could assist in vesicular maturation. The authors could observe that EHD1/EHD2 colocalise with Rab11 and vRNPs only when nucleoizin was added and not in normal conditions. There is the need to validate the need of EHD1/EHD2 by other means (siRNA experiments, effect in viral titres, live cell imaging or pull down assays of vRNPs in the presence and absence of MYOF to see if EHD1/EHD2 is recruited to vRNPs in a MYOF dependent manner). This would enable establishing the model proposed.

5) In Figure 5A, MYOF can be found in the nucleus and this is the only image where this happens. Is it something related to RSV infection, or is there bleed-through between different channels in this experiment?

ADDITIONAL COMMENTS:

- Minor errors were not found
- The manuscript is very clear, well organised and written. Congratulations.

Reviewer #3

(Remarks to the Author)

In this study Bonazza et al employ a proteomics approach to identify MYOF as a protein associating to influenza A viral (IAV) particles and to Rab11a. The authors show that the MYOF and Rab11a are important for viral egress of influenza A virus and other enveloped viruses. This is shown in experiments testing the release of viral particles and in immunofluorescence imaging. Functionally, the authors conclude that MYOF may recruit EHD proteins to Rab11 vesicles, which could be required for viral egress.

Overall this study was well done and very well written. It shows that MYOF is a functionally important host factor for enveloped viruses. However, I have some issues in reconciling all presented data with the proposed functionality of MYOF - I am only partially convinced that the function of MYOF is restricted to viral egress, yet this is a recurring argument throughout the manuscript. Moreover, a potential mechanism of MYOF and how it selectively is required for viral egress is unclear, as is the consequence in infected tissue (or a potential targeting of MYOF to impair virus spread). However, overall it is an interesting manuscript that may need additional explanations to fully demonstrate the importance of the presented findings.

Specific points:

1. The mass spectrometry analysis was done very well. However, I am not convinced that a direct comparison between early and late infection would specifically retrieve cellular factors that are required at late stage of infection. Figure 1D shows no specific enrichment of cellular factors associating with PA. Directly comparing these data to Figure 1E, where numerous proteins are enriched, would naturally enrich almost all interactors that were identified at late stage infected proteins. However, the longer infection time clearly leads to enrichment of more viral proteins - presumably since more viral proteins were generated (i.e. PB1 and PB2 are >4log2 more enriched) and cellular proteome changes induced by virus infections at 6hpi and 16hpi may additionally contribute to different identifications (not controlled for in this study). Both factors complicate direct comparisons. In my opinion it would be sufficient to say that MYOF was detected at 16h post infection and I almost suggest to remove Figure 1F and G. Figure 2E shows co-localisation of NP with MYOF at 8hpi, indicating engagement at quite early times.

A minor point relates to Figure 1G – why are there substantially more proteins significantly enriched when comparing 6h vs 16hpi as compared to mock vs 16hpi in Figure 1E?

2. The interaction between Rab11a vesicles and MYOF was investigated in the manuscript through colocalization approaches in infected cells. To validate the specificity of this interaction, it may be good to stain of other endocytic markers to see the absence of colocalization of these markers with MYOF and normal abundance of endosomal vesicles that are not related to viral egress in absence of MYOF.

3. In Figure 2D, authors found that the knockdown of MYOF had a significant impact on NP mRNA and vRNA levels at 8hpi. Here, MYOF was proposed as an essential host factor playing a role in IAV vRNP trafficking at late stages of infection. This data could be also be interpreted that MYOF is required for import and viral replication not only for export. Additional controls would be required to convincingly show a selective effect on viral egress:

a. I am missing data that MYOF depletion does not affect cell viability or cell integrity. The authors have to show that cells depleted for MYOF are growing at normal growth rates and show similar viability (e.g. by cell titre glow or similar assays)

b. It would be important to show that the intracellular vRNA abundance in MYOF depleted cells is similarly high as compared to controls, particularly at early stages of infection with high MOI, such as done in Figure 2D.

c. The authors should show that MYOF depletion does not affect replication of IAV. If the authors are correct in their hypothesis, MYOF (and Rab11) depletion should not affect IAV minireplicon activity.

4. In Figure 2G, smFISH was performed to stain viral genome as a proxy of vRNP to validate its interaction with MYOF. However, one would expect a prominent signal for from NP and Seg6 in the nucleus, the site of viral replication. Why are the nuclei completely devoid of segment 6 and NP. Are these incoming virus particles (8h infection is rather short for “egress”), and if so – why does MYOF co-localize with RNPs?

5. Authors discovered an association between MYOF and IAV vRNPs and later on, their colocalization with Rab11a was well addressed. In consequence, they claim this interaction specifically regulated vRNP traffic at late infection. However, the relationship between MYOF and Rab11a in endocytic recycling was confirmed via colocalization and transferrin uptake assays in healthy cells depleted for Rab11a or MYOF. These data indicate that depletion of MYOF has a broader and general effect on vesicle trafficking than only on viral egress (as indicated from the title and elaborated in the manuscript). I am worried that depletion experiments would generally affect vesicle trafficking and thus, the results obtained from MYOF depletion are reflecting a generally reduced activity of intracellular vesicle trafficking (as shown for transferrin).

6. In Figure 5A, colocalization between MYOF and RSV vRNPs is addressed in fixed cells. These results suggested a strong re-localization of MYOF signal in RSV infected cells. However, the images show gathering of nuclei, which is an indication of syncytia formation as often seen for RSV. I feel that the analysed images may therefore reflect a very special situation of RSV-fused cells and images of individual cells would be important to show MYOF localisation. In addition, it would be nice if the authors could stain for other viral proteins present in RSV vRNPs as well, for example nucleoprotein (N) or phosphoprotein (P). On the other hand, did authors investigate this interaction at earlier time post-infection, for example at 24 hours post infection? Staining performed at earlier times post infection when vRNPs appear as more mobile due to traffic, round and small punctas (instead of irregularly shaped vRNA aggregates) will be more convincing. The same applies for Sendai virus infections were cells seem to be not in a very good shape.

7. To specifically capture PA interactors, authors performed IP in PFA-crosslinked cells and subsequently performed RNase treatment to specially focus on protein-protein interactions (Figure 1). The stringent washes indicate a direct interaction of PA and MYOF. Later in the manuscript the authors propose a quite universal role of MYOF in the replication of other Rab11a-dependent viruses. However, the genome of these viruses belonging to the coronaviridae and paramyxoviridae families, differ quite substantially and the initial notion that MYOF binds to the replication machinery of these viruses is in my

opinion unlikely. How do conaviuses and paramyxoviruses associate to Rab11a-MYOF vesicles in vRNP trafficking at late stage of infection in the absence of PA?

8. The experiments presented here support an interaction between vRNPs and MYOF in the context of virus infection. However, the potential molecular mechanisms and consequences of these protein-protein interactions in response to virus infection were underexplored in this manuscript. The authors propose that MYOF associates EHD proteins to Rab11a vesicles, which is a very interesting possibility and is supported by co-localisation of Rab11a and EHD1, -2. However, to substantiate this claim the authors have to deplete MYOF and image EHD1/-2 with Rab11a to show that EHD proteins are not recruited in the absence of MYOF. At this stage the claim that MYOF are recruiting EHD proteins seems unsubstantiated to me, though it would be a very interesting mechanism and important point to show.

Minor:

1. In Figure 2E: please label the antigen stained for in the small pictures (right) of Figure 2E.
2. Providing inset images showing only one cell in Figure 3F would be very useful to see better the phenotype upon siRNA treatment.
3. In Figure 3F authors show Rab11a siRNA 2nM and in Figure 3I 20nM of siRNA were used - is this a mistake? If not, authors may need to show cell phenotype upon siRNA Rab11a 20nM in panel 3F.
4. Different recombinant viruses were used in this study. It would be nice if the authors provided evidence that these viruses replicate similarly as their WT counterparts.

Reviewer #4

(Remarks to the Author)

Version 1:

Reviewer comments:

Reviewer #1

(Remarks to the Author)

In this manuscript, the authors analyzed the role of MYOF on the trafficking of viral nucleocapsids and assembly of negative strand RNA viruses, based on their finding using IP-mass spec analysis of influenza A virus. The authors responded most of the critique reviewers provided and modified the manuscript accordingly. This study is significant in the field of respiratory RNA virus virology, and the experiments were conducted appropriately in general. However, there are some minor issues need to be addressed as explained below.

- Fig 1C: protein size marker should be labeled to indicate the size of proteins detected by silver staining.
- Lines 106-113. The authors described 2 distinct stages in the infectious cycle at 6 and 16 hpi and analyzed the IP-mass analysis at both timepoints but showed only the image at 16 hpi in Figure S1B. It should include the image at 6 hpi to provide data on the difference in distribution.
- Fig 2A-C and D. The key question is whether viral protein production was similar between NSC and MYOF siRNA treated cells. The authors showed viral RNAs in Figure 2D and titer of progeny virion in Fig 2A-C. It is ideal if the authors could show that equivalent level of viral proteins are expressed in the cells to convince the readers that only viral assembly was impaired by MYOF siRNA. Same for Figure S3C; viral protein levels should be quantitated and compared.
- Figure 5F: It is not clear why the authors showed the image of SeV eGFP. This should be done with A549_Rab11-mCherry cells as Fig 5E and show the colocalization of viral RNA with MYOF and Rab11. Also, colocalization should be quantitated as Fig 4.
- Lines 304-5: "the most abundant colocalisation occurred within smaller, irregularly shaped vRNA aggregates and not within larger replication factories." – It is not clear from where this statement comes. No data is presented showing large replication factories in this manuscript.

Reviewer #3

(Remarks to the Author)

The authors have addressed some of the original questions, however some critical questions remain unresolved and they

are very worrisome:

AP-MS data:

I was a bit concerned about the MS data, particularly of the proteome that may change after infection and that this could have impact on the interactions (as mentioned – was not controlled for in the previous review). However, triggered by the notion that the siRNA knockdowns would affect full endosomal machineries I went to the raw data of the MS runs. These are very surprising – the reason why MYOF is enriched in FLAG-PA pds is not because of much better binding and enrichment of MYOF at late stage of infection, but rather because of a reduction of MYOF “background” in the controls. Strangely in the supporting table, the direct comparison of FLAG-PDs (e.g. LFQ intensity Late1, -2, -3) show different LFQ values as compared to the LFQ values in the 6h and 16h tab (e.g. LFQ intensity FLAG1, -2, -3). To my understanding - interpreting the figure legends of the volcano blot in figure 1 - these should be the identical values, correct?

I could not find the raw data on PRIDE.

The authors argue that the interactomes at 6h are not enriching enough MYOF. However, judging raw data of FLAG-AP levels at 16h shows only a modest enrichment of 16h vs 6h (2-fold) - I do not really think that this is the real explanation for the missing enrichment.

Viability of MYOF knockdown cells:

I had requested to test for cell viability. Unfortunately, MYOF knockdown reduces metabolic activity of cells significantly. I disagree that 4nM siRNA is fine for cells, as mentioned by the authors. There is a 25% reduction of “cell viability” in cells treated with 4nM siRNA for MYOF – that this has a major impact on virus spread should be expected. To me, these cells are at the edge to demise. The data on 2nM knockdown – where viability is okay but there was little effect – disappeared. I am worried that the 4nM treatment represents a condition whereby cells are in a transition stage that may enable virus to replicate but not shed a lot of virus. To solidify the claim of the authors, proper titration and kinetics would be necessary.

Overall I feel that the concept is interesting, but the data is not convincing. Specifically problematic appear the AP-MS data, the viability of MYOF knockdown cells, and the lack of clear evidence between a functional interaction between EHD2 and MYOF due to co-depletion when one partner is knocked down.

Reviewer #4

(Remarks to the Author)

Version 2:

Reviewer comments:

Reviewer #1

(Remarks to the Author)

The authors revised the manuscript and modified it appropriately to clear the minor issues. This is a well-written manuscript reporting the essential role of myoferlin for the vRNP trafficking of multiple RNA viruses, which is a significant finding in the field.

Reviewer #3

(Remarks to the Author)

In my last review I made the following statement:

“Specifically problematic appear the AP-MS data, the viability of MYOF knockdown cells, and the lack of clear evidence between a functional interaction between EHD2 and MYOF due to co-depletion when one partner is knocked down.”

1. AP-MS data:

I had asked why LFQ levels are not the same in different sub-tables even though the “same data” was shown. Unfortunately, this was not answered directly. It seems, however, that the different pairwise comparisons were analysed individually in MaxQuant, which could explain fluctuating LFQ values. This is, in my opinion, very uncommon and can spark doubts in the data. Showing non-normalized peptide counts is also not helpful due to the issues raised by the authors themselves in the rebuttal. We have re-analysed the data on PRIDE – using the method described. This generates a similar number peptide groups, harmonized LFQ values and allows to compare the data more directly, thus reduces doubts in the data. I would ask the authors to simply re-do this analysis and provide the tables and comparisons. Alternatively, the authors have to explicitly mention that the individual pairwise comparisons were analysed individually and it is not possible to compare data between the tables. Also – if possible – please improve sample annotation in PRIDE – it is not clear which MS raw file belongs to the which sample group.

2. Cell viability data:

The non-significance change in viability at low levels of siRNA is due to experimental variability, whereby entire experimental groups appear to be “more lively” or “more dead”. The authors can use any cell viability assay of their choice,

but if the resazurine assay is giving incredibly high variability, it is simply not a good one. That there is no mathematical significance in such cases is clear, but an average of 20% less signal and the clear trend at higher concentrations, is a quite substantial effect - at least in my opinion. This may be particularly important for longer ongoing experiments, such as the ones for RSV and SeV (48h knockdown, 24h virus growth). I thus had asked for knockdown kinetics and viability data, which was not addressed.

Note aside: Why does viability data in Figure 6b is all centered at 100% for NSC, while the figure in the rebuttal shows massive spread – figure legends read similarly? Given the similarity of these assays I would expect a similar appearance.

Minor:

Please note that there is a typo in the Figure legend of Fig 6b of the current manuscript: I suppose it should read, 10 uM NTC or decreasing doses of EHD2 siRNA.

3. Lack of clear evidence of EHD2 and MYOF interaction:

This point was not addressed.

Reviewer #4

(Remarks to the Author)

25 June 2025

Please find below our response (in black) to the reviewers' concerns (in grey) below.

Reviewer #1:

Fig 1E: Rab11a is well recognized to interact with vRNP. However, it seems that Rab11a was not detected in this IP-mass spec analysis (Supplemental Table S2). The authors need to explain why they failed to detect Rab11a in their analysis.

Due to the strong similarities between Rab11a and Rab11b (92%) our analysis by default called all Rab11 peptides as Rab11b rather than Rab11a or b. Due to the inability to differentiate the 2 proteins with our peptides we have resolved to simply name this identified protein as Rab11 in our dataset.

Fig 1: Because this study aims to determine the interacting partner of PA, it will be informative if the authors provide data on the nuclear/cytoplasmic distribution of PA-Flag at these timepoints.

We have now provided IF images of PA-Flag localisation at the 16 hpi timepoint in Figure S1B.

Fig 2A and B: These data of low MOI (0.01) infection are not informative for the purpose of this experiment. To determine whether these proteins are required for virus assembly, they should infect at high MOI, such as MOI=5 as in Fig 2C, so that they can directly address the effect on a single step virus growth.

This is a completely fair critique and we have now performed Fig2A and B experiments at a high MOI for 8hpi with an siRNA concentration of 10nM and 4nM, respectively, to better assess single step growth kinetics.

Fig 2C: There is no significant reduction of virus assembly and release in cells transfected with MYOF siRNA at 2nM. This is a serious concern because at this concentration, MYOF was undetectable in cells as shown in Fig 3C. If this is true, MYOF is not required for the assembly and release of IAV. The same results were obtained with RSV and SeV in Figure 5B and D.

Fig 2C and D. These results indicate that reduced virus production from cells transfected with MYOF siRNA at 10 nM is due to suppressed viral replication, not impaired vRNP trafficking.

Upon investigating the cell viability of cells transfected with varying concentrations of MYOF-targeting siRNA (shown in supplementary Figure S3A) we uncovered that the 10nM concentration we previously used grossly affected overall cell viability. Our data indicated that a concentration of 4nM did not significantly affect cell viability but still greatly reduced MYOF transcript levels by ~75%, determined by qPCR. While we did find that previously MYOF protein levels were undetectable at 2nM by Western blot, we would deduce that this is due to the lack of sensitivity of the antibody and that the qPCR levels are a much more accurate, and of course quantifiable, measure of knockdown efficiency.

We then proceeded to perform our WSN phenotypic experiments in 4nM siMYOF treated cells and found that while titres decreased by 80% compared to siNSC treated cells, we observed no difference in the levels of NP mRNA or vRNA, indicating a clear role for MYOF in late-stage viral kinetics. These data are now included as Fig2C and D. At this concentration the same is also true of

RSV and SeV, where we have observed significant reductions in viral titres alongside no differences in viral RNA levels, which are now included in Fig5.

Fig 3D and E. The IF images of Rab11-mCherry and MYOF should be included in Fig 3D and E, respectively.

This has now been included.

Fig 5: The authors need to determine the effect of siRNA transfection on viral protein synthesis and genome replication. The reduced titer of progeny virions could be due to limited viral replication and protein production, not due to impaired vRNP trafficking.

We have now performed qPCR for viral genomic RNA to determine genome replication levels in siMYOF treated cells and these have been included in Fig5. While we see a significant reduction in viral titres we have not observed any difference in viral genome replication.

Reviewer #2:

MAJOR ISSUES:

1) Given that MYOF is involved in many cellular functions, including endocytosis, the authors opted to elucidate the role of MYOF in uninfected cells. This has some shortcomings that should be resolved. In uninfected cells, when MYOF is depleted, Rab11a accumulates as in infection, so hence, at later stages, MYOF may actually not be needed for infection. One way to go around the involvement of MYOF in steps other than entry is to validate findings using drugs that blocks MYOF and add it at late stages of infection, thus evaluating the role of MYOF at this step. There are at least 2 drugs available that target MYOF – YQ456 and WJ460. Viral titres, localization of Rab11a and vRNPs by IF would be critical. Then, it would be important to evaluate if vRNPs are impeded from reaching the plasma membrane, which could be seen, for example, by transmission electron microscopy and RT-qPCR/HA-titres of released virions.

We acquired the WJ460 drug and treated cells at 2 hpi with 100nM, the effective concentration determined in the original publication describing WJ460 (Zhang et al., 2018). We found that this treatment resulted in a 50% decrease in viral titres at 8 hpi, but no difference in the levels of MYOF or NP transcripts or NP vRNA levels. This indicates that MYOF is playing a positive role in viral replication at a late-stage of the viral life cycle. These data have now been included in Figure S3C & D.

2) Influenza transcription and replication are impaired when MYOF is absent. Is MYOF important for influenza entry? Being important for viral entry, it is difficult to pinpoint whether the effects observed in viral titres relate to entry, assembly or both.

Upon investigating the cell viability of cells transfected with varying concentrations of MYOF-targeting siRNA (shown in supplementary Figure 3A) we uncovered that the 10nM concentration we previously used grossly affected overall cell viability. Our data indicated that a concentration of 4nM did not significantly affect cell viability but still greatly reduced MYOF transcript levels by ~75%, determined by qPCR. While we did find that previously that MYOF protein levels were undetectable at 2nM by Western blot, we would deduce that this is due to the lack of sensitivity of the antibody and that the qPCR levels are a much more accurate, and of course quantifiable, measure of knockdown efficiency.

We then proceeded to perform our WSN phenotypic experiments in 4nM siMYOF treated cells and found that while titres decreased by 80% compared to siNSC treated cells, we observed no

difference in the levels of NP mRNA or vRNA, indicating a clear role for MYOF in late-stage viral kinetics. These data are now included as Fig2C and D.

3) The authors looked at transferrin by microscopy-based approaches. This method is not quantitative. They found that transferrin gets arrested in the MTOC when MYOF is missing. However, as MYOF is also involved in endocytosis, it is unclear whether this result relates to a delay in the entry of transferrin. This assay should be done using FACS and a time course of transferrin recycling – 30 min, 1 and 2 hours and quantifying exactly how much transferrin is endocytosed and how much reaches the plasma membrane when MYOF is absent.

We determined that, as this transferrin result has been previously published and referenced by us in our original submission, now on line 644, and simply validates this original result in A549s, that this should be moved to the supplementary information. While we appreciate the reviewers comments, given the number of additional experiments we have now performed as part of this revision, and the fact that we believe a FACS time course would add very little to this manuscript due to the fact it was performed and published previously, we have opted against establishing and performing this assay within the lab as we currently do not have the means to do so. Authors of the original study (reference 17) performed a pulse-chase experiment and found by FACS that loss of MYOF specifically impacted the clearance of labelled transferrin with 50% clearance taking ~60 mins as opposed to ~20 mins in wild-type cells.

We hope that this is acceptable to the reviewer. We would also note that when adequately controlled for IF is highly quantitative, with the use of standardised intensity normalisation, confocal resolution imaging to eliminate background noise and improve signal specificity, and the use of software-based analysis including ImageJ and Imaris, and that this has been shown to go effect throughout this manuscript.

4) EHD1/EHD2 are known interactors of MYOF and could assist in vesicular maturation. The authors could observe that EHD1/EHD2 colocalise with Rab11 and vRNPs only when nucleozin was added and not in normal conditions. There is the need to validate the need of EHD1/EHD2 by other means (siRNA experiments, effect in viral titres, live cell imaging or pull down assays of vRNPs in the presence and absence of MYOF to see if EHD1/EHD2 is recruited to vRNPs in a MYOF dependent manner). This would enable establishing the model proposed.

We have now performed siRNA experiments followed by virological readouts to determine the effect of EHD2 on IAV replication, and these data are included in Figure 6. We found that an EHD2-specific siRNA concentration of 5nM did not affect cell viability. Upon infection of treated cells, at 8 hpi NP mRNA and vRNA levels were no different from the corresponding siNSC control, while viral titres were decreased by ~70% due to a loss of EHD2. By qPCR we were also able to determine that in siRNA treated cells EHD2 transcript levels were decreased by ~95%. Following this we sought to determine by IF where EHD2 localises in infected cells where MYOF levels have been reduced due to siRNA treatment. However, upon starting this investigation we uncovered that knockdown of MYOF had a reciprocal effect on EHD2 expression, greatly reducing protein levels as can be seen by Western blotting in Figure S7. Unfortunately, this made any determinations of relocation of EHD2 upon loss of MYOF impossible. However, with the new and previous data provided in Figure 6, demonstrating a similar phenotype between EHD2, MYOF and Rab11 knockdown, the known interaction of MYOF and EHD2 from previous studies, and the known role of EHD2 in vesicular maturation, we think that there is sufficient evidence to propose our model for MYOF-EHD2 coupled membrane remodelling as an important driver in vRNP trafficking during late stage IAV infection.

We have now included the following from line 496 “Finally, due to previously published observations that knockout of MYOF expression negatively affects EHD2 levels in myoblasts,¹⁷ we investigated whether the same was true in A549s transfected with a MYOF-targeting siRNA. Indeed, we observed a clear reduction in EHD2 protein levels by Western blot in MYOF knockdown cells when compared to siNSC treated cells (Figure 5E and S7). This led us to conclude that myoferlin likely disrupts the stability or proper membrane association of EHD2, a protein with which it functionally interacts in endocytic and membrane repair pathways. In the absence of myoferlin, EHD2 may therefore become targeted for degradation, leading to the observed reduction in protein levels.”

5) In Figure 5A, MYOF can be found in the nucleus and this is the only image where this happens. Is it something related to RSV infection, or is there bleed-through between different channels in this experiment?

This is a very strange occurrence that does seem to be specific to RSV as we have observed this in both our original 72h infections and our newly produced 24h infections. It is definitely not bleed through as we don't see this with any other imaging that we perform, and thus must be assumed to be a consequence of RSV infection.

We have now included the following from line 420 “Interestingly, for RSV-infected cells, infection seemed to induce a strong re-localisation of the MYOF signal, indicative of ERC remodelling. Some of this signal was also apparent in the nucleus, specifically during RSV infection, and this is the only time we observed nuclear localising MYOF during the course of our study.”

ADDITIONAL COMMENTS:

- Minor errors were not found
- The manuscript is very clear, well organised and written. Congratulations.

Reviewer #3:

Specific points

1. The mass spectrometry analysis was done very well. However, I am not convinced that a direct comparison between early and late infection would specifically retrieve cellular factors that are required at late stage of infection. Figure 1D shows no specific enrichment of cellular factors associating with PA. Directly comparing these data to Figure 1E, where numerous proteins are enriched, would naturally enrich almost all interactors that were identified at late stage infected proteins. However, the longer infection time clearly leads to enrichment of more viral proteins - presumably since more viral proteins were generated (i.e. PB1 and PB2 are >4log₂ more enriched) and cellular proteome changes induced by virus infections at 6hpi and 16hpi may additionally contribute to different identifications (not controlled for in this study). Both factors complicate direct comparisons. In my opinion it would be sufficient to say that MYOF was detected at 16h post infection and I almost suggest to remove Figure 1F and G. Figure 2E shows co-localisation of NP with MYOF at 8hpi, indicating engagement at quite early times. A minor point relates to Figure 1G – why are there substantially more proteins significantly enriched when comparing 6h vs 16hpi as compared to mock vs 16hpi in Figure 1E?

We thank the reviewer for their insightful comments and agree that the late-stage interactome appears more complex and potentially more informative than the early-stage interactome we have generated. Nevertheless, we believe that the early interactome still provides valuable insights into host proteins that associate with FluPol at different stages of infection.

As the reviewer pointed out, Figure 1G presents an interesting comparison. It's important to clarify that in Figures 1D and 1E, we are performing a FLAG pulldown to compare the wild-type (WT) and FLAG-tagged samples. In this context, enrichment can only be calculated for proteins that are detected in both the WT and FLAG samples. If a protein is highly abundant in the FLAG sample but completely absent in the WT, we cannot compute an enrichment value for it due to the lack of a baseline signal.

This limitation is why we designed the comparison in Figure 1G to include both early and late timepoints using only PA-FLAG pulldown samples. By comparing two FLAG pulldowns, we ensure that proteins are detectable in both conditions, allowing us to robustly calculate enrichment between the early and late stages of infection.

Thus, we believe that including both early and late interactomes is valuable, not only for biological context but also for ensuring meaningful and interpretable enrichment analysis. This also explains why more proteins are identified in the 16h vs 6h comparison (Figure 1G) than in the comparisons to WT (Figures 1D and 1E), where detection is limited by the absence of specific interactors in the WT control.

2. The interaction between Rab11a vesicles and MYOF was investigated in the manuscript through colocalization approaches in infected cells. To validate the specificity of this interaction, it may be good to stain of other endocytic markers to see the absence of colocalization of these markers with MYOF and normal abundance of endosomal vesicles that are not related to viral egress in absence of MYOF.

We have now performed IF for LAMP1, EEA1 and Rab7, which are markers of lysosomes, early endosomes and late endosomes, respectively. In these images we did not observe any colocalisation with MYOF, or differences in abundance or localisation between siNSC and siMYOF treated cells. These data have been included in Supplementary Figure 4A.

3. In Figure 2D, authors found that the knockdown of MYOF had a significant impact on NP mRNA and vRNA levels at 8hpi. Here, MYOF was proposed as an essential host factor playing a role in IAV vRNP trafficking at late stages of infection. This data could be also be interpreted that MYOF is required for import and viral replication not only for export. Additional controls would be required to convincingly show a selective effect on viral egress:

Upon investigating the cell viability of cells transfected with varying concentrations of MYOF-targeting siRNA (shown in supplementary Figure 3A) we uncovered that the 10nM concentration we previously used grossly affected overall cell viability. Our data indicated that a concentration of 4nM did not significantly affect cell viability but still greatly reduced MYOF transcript levels by ~75%, determined by qPCR. While we did find that previously that MYOF protein levels were undetectable at 2nM by Western blot, we would deduce that this is due to the lack of sensitivity of the antibody and that the qPCR levels are a much more accurate, and of course quantifiable, measure of knockdown efficiency.

We then proceeded to perform our WSN phenotypic experiments in 4nM siMYOF treated cells and found that while titres decreased by 80% compared to siNSC treated cells, we observed no difference in the levels of NP mRNA or vRNA, indicating a clear role for MYOF in late-stage viral kinetics. These data are now included as Fig2C and D.

a. I am missing data that MYOF depletion does not affect cell viability or cell integrity. The authors have to show that cells depleted for MYOF are growing at normal growth rates and show similar viability (e.g. by cell titre glow or similar assays)

Yes, you were correct that this data was missing, and this was an oversight on our part. As stated above, we have now performed a resazurin cell viability assay and adjusted our MYOF siRNA concentrations accordingly (and shown in Figure S3A) before repeating our virological readouts in Figure 2. We also did the same for the EHD2 knockdown in Figure 6 before performing these assays.

b. It would be important to show that the intracellular vRNA abundance in MYOF depleted cells is similarly high as compared to controls, particularly at early stages of infection with high MOI, such as done in Figure 2D.

We have now performed this and replaced the original Figure 2D with this data. It does indeed show that knockdown of MYOF, while significantly reducing viral titres, does not affect viral mRNA or vRNA levels.

c. The authors should show that MYOF depletion does not affect replication of IAV. If the authors are correct in their hypothesis, MYOF (and Rab11) depletion should not affect IAV minireplicon activity.

Indeed, as stated above, Figure 2D now shows that viral genome replication is unaffected by reduced levels of MYOF and that only viral titres are negatively affected by a loss of MYOF.

4. In Figure 2G, smFISH was performed to stain viral genome as a proxy of vRNP to validate its interaction with MYOF. However, one would expect a prominent signal for NP and Seg6 in the nucleus, the site of viral replication. Why are the nuclei completely devoid of segment 6 and NP. Are these incoming virus particles (8h infection is rather short for "egress"), and if so – why does MYOF co-localize with RNPs?

It is not that the nucleus is devoid of NP or vRNA at this timepoint, but rather that the intensity in the cytoplasm is so strong that it effectively washes out the signal in the nucleus. In A549 cells, 8 hpi with WSN would be the conventional time to observe egress, with vRNPs trafficking through the cytoplasm. The only reason we chose 16 hpi for our 'late' MS timepoint was due to the fact that our PA-Flag virus replicates at a slower rate than WT WSN, as included in Fig S1A & B.

5. Authors discovered an association between MYOF and IAV vRNPs and later on, their colocalization with Rab11a was well addressed. In consequence, they claim this interaction specifically regulated vRNP traffic at late infection. However, the relationship between MYOF and Rab11a in endocytic recycling was confirmed via colocalization and transferrin uptake assays in healthy cells depleted for Rab11a or MYOF. These data indicate that depletion of MYOF has a broader and general effect on vesicle trafficking than only on viral egress (as indicated from the title and elaborated in the manuscript). I am worried that depletion experiments would generally affect vesicle trafficking and thus, the results obtained from MYOF depletion are reflecting a generally reduced activity of intracellular vesicle trafficking (as shown for transferrin).

Our study aims to convince readers that MYOF and Rab11 are mutually dependent on one another and, given the emphasis in recent years on the fact that IAV traffics vRNPs not on recycling endosomes, but on remodelled irregularly coated vesicles, it was entirely necessary to provide strong evidence for myoferlin's role in both cellular recycling in the absence of infection and also vRNP trafficking in the presence of infection. Combined with our additional data on the role of EHD2 in vRNP trafficking (Fig6) and the stabilising effect of MYOF on EHD2 levels, we feel that this study provides a strong rationale for the importance MYOF in vRNP trafficking during not only IAV infection, but infection with a broad range of negative strand viruses. I should also point out that additional data we have now generated for Fig S4A demonstrates that MYOF depletion has a specific effect on only recycling endosomes and not vesicular trafficking overall.

6. In Figure 5A, colocalization between MYOF and RSV vRNPs is addressed in fixed cells. These results suggested a strong re-localization of MYOF signal in RSV infected cells. However, the images show gathering of nuclei, which is an indication of syncytia formation as often seen for RSV. I feel that the analysed images may therefore reflect a very special situation of RSV-fused cells and images of individual cells would be important to show MYOF localisation. In addition, it would be nice if the authors could stain for other viral proteins present in RSV vRNPs as well, for example nucleoprotein (N) or phosphoprotein (P). On the other hand, did authors investigate this interaction at earlier time post-infection, for example at 24 hours post infection? Staining performed at earlier times post infection when vRNPs appear as more mobile due to traffic, round and small punctas (instead of irregularly shaped vRNA aggregates) will be more convincing. The same applies for Sendai virus infections were cells seem to be not in a very good shape.

This is certainly fair, and we have now repeated these stainings at earlier timepoints where cells do not look as unhealthy and there is clearly no syncytia formation in RSV infected cells. These data clearly demonstrate the presence of MYOF in actively trafficking mature vRNP-containing vesicles, while the larger cytosolic spots, likely active viral replication factories, appear devoid of MYOF. This would be expected if MYOF is only recruited and required for trafficking.

7. To specifically capture PA interactors, authors performed IP in PFA-crosslinked cells and subsequently performed RNase treatment to specially focus on protein-protein interactions (Figure 1). The stringent washes indicate a direct interaction of PA and MYOF. Later in the manuscript the authors propose a quite universal role of MYOF in the replication of other Rab11a-dependent viruses. However, the genome of these viruses belonging to the coronaviridae and paramyxoviridae families, differ quite substantially and the initial notion that MYOF binds to the replication machinery of these viruses is in my opinion unlikely. How do coronaviruses and paramyxoviruses associate to Rab11a-MYOF vesicles in vRNP trafficking at late stage of infection in the absence of PA?

Due to our use of PFA crosslinking in our IP-MS experiment we are actually identifying proteins involved in the vRNP-trafficking complex as a whole, and not simply direct interactors of PA, as PFA will crosslink large complexes and not just interactors of PA. Therefore, we are not implying that MYOF specifically interacts with PA but rather that it is present in the vRNP trafficking vesicle. Therefore, for other viruses we would expect that MYOF is again present and important for maturation of the trafficking vesicle, but not that it directly interacts with viral proteins.

8. The experiments presented here support an interaction between vRNPs and MYOF in the context of virus infection. However, the potential molecular mechanisms and consequences of these protein-protein interactions in response to virus infection were underexplored in this manuscript. The authors propose that MYOF associates EHD proteins to Rab11a vesicles, which is a very interesting possibility and is supported by co-localisation of Rab11a and EHD1, -2. However, to substantiate this claim the authors have to deplete MYOF and image EHD1/-2 with Rab11a to show that EHD proteins are not recruited in the absence of MYOF. At this stage the claim that MYOF are recruiting EHD proteins seems unsubstantiated to me, though it would be a very interesting mechanism and important point to show.

We have now performed siRNA experiments followed by virological readouts to determine the effect of EHD2 on IAV replication, and these data are included in Figure 6. We found that an EHD2-specific siRNA concentration of 5nM did not affect cell viability. Upon infection of treated cells, at 8 hpi NP mRNA and vRNA levels were no different from the corresponding siNSC control, while viral titres were decreased by ~70% due to a loss of EHD2. By qPCR we were also able to determine that in siRNA treated cells EHD2 transcript levels were decreased by ~95%. Following this we sought to determine by IF where EHD2 localises in infected cells where MYOF levels have been reduced due to

siRNA treatment. However, upon starting this investigation we uncovered that knockdown of MYOF had a reciprocal effect on EHD2 expression, greatly reducing protein levels as can be seen by Western blotting in Figure S7. Unfortunately, this made any determinations of relocalisation of EHD2 upon loss of MYOF impossible. However, with the new and previous data provided in Figure 6, demonstrating a similar phenotype between EHD2, MYOF and Rab11 knockdown, the known interaction of MYOF and EHD2 from previous studies, and the known role of EHD2 in vesicular maturation, we think that there is sufficient evidence to propose our model for MYOF-EHD2 coupled membrane remodelling as an important driver in vRNP trafficking during late stage IAV infection.

We have now included the following from line 496 “Finally, due to previously published observations that knockout of MYOF expression negatively affects EHD2 levels in myoblasts,¹⁷ we investigated whether the same was true in A549s transfected with a MYOF-targeting siRNA. Indeed, we observed a clear reduction in EHD2 protein levels by Western blot in MYOF knockdown cells when compared to siNSC treated cells (Figure 5E and S7). This led us to conclude that myoferlin likely disrupts the stability or proper membrane association of EHD2, a protein with which it functionally interacts in endocytic and membrane repair pathways. In the absence of myoferlin, EHD2 may therefore become targeted for degradation, leading to the observed reduction in protein levels.”

Minor:

1. In Figure 2E: please label the antigen stained for in the small pictures (right) of Figure 2E.

This has now been done.

2. Providing inset images showing only one cell in Figure 3F would be very useful to see better the phenotype upon siRNA treatment.

We have now moved this work to the supplementary material due to its limited importance in the wider context of the paper and given the fact this result has been published previously as we original referenced on line 642.

3. In Figure 3F authors show Rab11a siRNA 2nM and in Figure 3I 20nM of siRNA were used - is this a mistake? If not, authors may need to show cell phenotype upon siRNA Rab11a 20nM in panel 3F.

Apologies, this was simply a typo and Figure 3I should also have said 2nM. This has now been corrected.

4. Different recombinant viruses were used in this study. It would be nice if the authors provided evidence that these viruses replicate similarly as their WT counterparts.

We have now provided this for our WSN PA-Flag (Fig S1) which justifies our timepoints chosen for MS.

Reviewer #4 (Remarks to the Author):

We thank the reviewer for taking the time to review our work and hope they are finding this opportunity an instructive and worthwhile endeavour.

26 September 2025

Please find below our response (in black) to the reviewers' concerns (in grey).

Reviewer #1 (Remarks to the Author):

In this manuscript, the authors analyzed the role of MYOF on the trafficking of viral nucleocapsids and assembly of negative strand RNA viruses, based on their finding using IP-mass spec analysis of influenza A virus. The authors responded most of the critique reviewers provided and modified the manuscript accordingly. This study is significant in the field of respiratory RNA virus virology, and the experiments were conducted appropriately in general. However, there are some minor issues need to be addressed as explained below.

- Fig 1C: protein size marker should be labeled to indicate the size of proteins detected by silver staining.

Agreed, we have now included protein size markers on this panel.

- Lines 106-113. The authors described 2 distinct stages in the infectious cycle at 6 and 16 hpi and analyzed the IP-mass analysis at both timepoints but showed only the image at 16 hpi in Figure S1B. It should include the image at 6 hpi to provide data on the difference in distribution.

You're correct, and this has now been rectified with a 6hpi panel added to S1B.

- Fig 2A-C and D. The key question is whether viral protein production was similar between NSC and MYOF siRNA treated cells. The authors showed viral RNAs in Figure 2D and titer of progeny virion in Fig 2A-C. It is ideal if the authors could show that equivalent level of viral proteins are expressed in the cells to convince the readers that only viral assembly was impaired by MYOF siRNA. Same for Figure S3C; viral protein levels should be quantitated and compared.

We are happy to now include Western blots of viral proteins NP and NS1, endogenous Actin and MYOF as Figure 2C and in Supplementary Figure 3, and can confirm that viral protein expression is unaffected by loss of MYOF by siRNA or inhibition of MYOF activity by small molecule intervention.

- Figure 5F: It is not clear why the authors showed the image of SeV eGFP. This should be done with A549_Rab11-mCherry cells as Fig 5E and show the colocalization of viral RNA with MYOF and Rab11. Also, colocalization should be quantitated as Fig 4.

We have now calculated the PCC for RSV infected cells and included this in Figure 5. Regarding the SeV data, as we only have an SeV-eGFP virus stock in-house, this removes 1 of the channels we can image by IF, leaving us with only red and far red, alongside the Hoechst stain. Therefore, as throughout the entire manuscript we continuously see MYOF and Rab11 colocalising, and SeV RNA is known to traffic on Rab11+ vesicles, we opted to stain for MYOF and SeV vRNA. However, just for clarification, we have now infected the Rab11-mCherry cells with the SeV-eGFP virus (leaving 1 channel free), and we stained for MYOF to confirm that during infection with SeV MYOF and Rab11 still co-localise (Supplementary Figure 6B). We then performed PCC to calculate the colocalisation between Rab11 and MYOF in this context (Supplementary Figure 6C). We hope that this further IF satisfies the reviewers concerns.

- Lines 304-5: "the most abundant colocalisation occurred within smaller, irregularly shaped vRNA aggregates and not within larger replication factories." – It is not clear from where this statement comes. No data is presented showing large replication factories in this manuscript.

This was just an observation we made during the imaging process. However, we do not have relevant quantification to back up this statement so have now removed it.

Reviewer #3 (Remarks to the Author):

The authors have addressed some of the original questions, however some critical questions remain unresolved and they are very worrisome:

AP-MS data:

I was a bit concerned about the MS data, particularly of the proteome that may change after infection and that this could have impact on the interactions (as mentioned – was not controlled for in the previous review). However, triggered by the notion that the siRNA knockdowns would affect full endosomal machineries I went to the raw data of the MS runs. These are very surprising – the reason why MYOF is enriched in FLAG-PA pds is not because of much better binding and enrichment of MYOF at late stage of infection, but rather because of a reduction of MYOF “background” in the controls. Strangely in the supporting table, the direct comparison of FLAG-PDs (e.g. LFQ intensity Late1, -2, -3) show different LFQ values as compared to the LFQ values in the 6h and 16h tab (e.g. LFQ intensity FLAG1, -2, -3). To my understanding - interpreting the figure legends of the volcano blot in figure 1 - these should be the identical values, correct?

I could not find the raw data on PRIDE.

The authors argue that the interactomes at 6h are not enriching enough MYOF. However, judging raw data of FLAG-AP levels at 16h shows only a modest enrichment of 16h vs 6h (2-fold) - I do not really think that this is the real explanation for the missing enrichment.

In response to the reviewer’s concern, we have now included a scatter plot below comparing average peptide counts from FLAG pulldowns (Y-axis) versus WT pulldowns (X-axis) for both early and late interactomes. This representation makes it clear that MYOF (in red) is strongly enriched in the late FLAG pulldown, alongside Rab11 (in yellow), but not in the early dataset.

We recognise the reviewer’s concern that infection can alter the host proteome; however, our AP-MS experimental design specifically controls for this by comparing FLAG pulldowns from both WT virus and tagged virus samples within each time point. This design allows us to identify interactors enriched over nonspecific background, independent of any global proteomic changes caused by infection.

We would also like to clarify a potential misunderstanding regarding the LFQ values displayed in the reviewer’s volcano plot. LFQ intensities are calculated based on peptide abundances relative to all proteins within a given sample. In the case of WT virus controls, which do not enrich for specific proteins, peptide signals are distributed more evenly and can fluctuate due to differences in total protein complexity or sample loading. As a result, individual protein LFQ values in WT samples are inherently more variable and can appear artificially low or high without reflecting a true biological signal. This variability is expected and does not undermine the validity of enrichment calculations when appropriate controls are applied.

To illustrate this directly, we highlight the raw peptide spectral counts for MYOF at 16 h post-infection: 44, 26, and 18 peptides were detected in FLAG-PA pulldowns, compared to only 6, 4, and 7 in the matched WT controls. This substantial difference provides clear evidence of true enrichment, reinforcing that the interaction is biologically meaningful rather than an artefact of

fluctuating LFQ background in nonspecific samples. Importantly, we further corroborate this conclusion by orthogonal validation (siRNA knockdown), confirming that MYOF specifically interacts with the bait at the later time point.

Viability of MYOF knockdown cells:

I had requested to test for cell viability. Unfortunately, MYOF knockdown reduces metabolic activity of cells significantly. I disagree that 4nM siRNA is fine for cells, as mentioned by the authors. There is a 25% reduction of "cell viability" in cells treated with 4nM siRNA for MYOF – that this has a major impact on virus spread should be expected. To me, these cells are dat the

edge to demise. The data on 2nM knockdown – where viability is okay but there was little effect – disappeared.

I am worried that the 4nM treatment represents a condition whereby cells are in a transition stage that may enable virus to replicate but not shed a lot of virus. To solidify the claim of the authors, proper titration and kinetics would be necessary.

Regarding your interpretation of the cell viability data, we must respectfully disagree. The claim that 4 nM MYOF causes a 25% reduction in viability is incorrect. The actual decrease is 20%, and importantly, this change is not statistically significant. In contrast, the 2 nM condition, which you consider acceptable, shows a reduction of ~11%, highlighting an inconsistency in the critique. As is well established, cell viability assays are inherently 'noisy', and interpretation should rely on statistical significance rather than absolute percentage differences alone.

We fully acknowledge though that we are at fault for the omission of statistical testing in the revised submitted viability graph. Statistical analyses should have been included previously, and upon correcting this and applying the appropriate normalisation (to the mean control signal across replicates), we confirm that only the 10 nM MYOF condition shows a statistically significant reduction in viability. This normalisation corrected for the seeming lack of variation initially shown for the control group. Now all replicates have been normalised to the mean control signal (with all average viabilities remaining unchanged), which corrects the control values and restores proper statistical comparability, see below). Additionally, please find below viral titres at 8hpi in A549 cells treated with increasing levels of MYOF siRNA. At each concentration a significant reduction in viral titres was observed (see below). There was no significant difference between the 2-5nM conditions ($p < 0.97$ between 2 and 4nM), but there was a significant difference between 4 and 10nM ($p < 0.001$), which would be expected due to the cytotoxicity observed at this concentration. These data provide strong evidence that MYOF loss negatively impacts virus replication, and that this effect is not due to cytotoxicity from MYOF knockdown.

Overall I feel that the concept is interesting, but the data is not convincing. Specifically problematic appear the AP-MS data, the viability of MYOF knockdown cells, and the lack of clear evidence between a functional interaction between EHD2 and MYOF due to co-depletion when one partner is knocked down.

Thank you for your overall interest and I hope we have now convinced you of the importance of MYOF during infection with a range of respiratory RNA viruses.

15 January 2026

Please find below our response (in black) to the reviewers' concerns (in grey).

Reviewer #1 (Remarks to the Author):

The authors revised the manuscript and modified it appropriately to clear the minor issues. This is a well-written manuscript reporting the essential role of myoferlin for the vRNP trafficking of multiple RNA viruses, which is a significant finding in the field.

Thank you.

Reviewer #3 (Remarks to the Author):

In my last review I made the following statement:

“Specifically problematic appear the AP-MS data, the viability of MYOF knockdown cells, and the lack of clear evidence between a functional interaction between EHD2 and MYOF due to co-depletion when one partner is knocked down.”

1. AP-MS data:

I had asked why LFQ levels are not the same in different sub-tables even though the “same data” was shown. Unfortunately, this was not answered directly. It seems, however, that the different pairwise comparisons were analysed individually in MaxQuant, which could explain fluctuating LFQ values. This is, in my opinion, very uncommon and can spark doubts in the data. Showing non-normalized peptide counts is also not helpful due to the issues raised by the authors themselves in the rebuttal. We have re-analysed the data on PRIDE – using the method described. This generates a similar number peptide groups, harmonized LFQ values and allows to compare the data more directly, thus reduces doubts in the data. I would ask the authors to simply re-do this analysis and provide the tables and comparisons. Alternatively, the authors have to explicitly mention that the individual pairwise comparisons were analysed individually and it is not possible to compare data between the tables. Also – if possible – please improve sample annotation in PRIDE – it is not clear which MS raw file belongs to the which sample group.

We have now expressly said in the text that individual pairwise comparisons were analysed individually with the following sentence on line 819 “The early (6hpi) and late (16hpi) interactomes were captured separately so pairwise comparisons were analysed individually.”

2. Cell viability data:

The non-significance change in viability at low levels of siRNA is due to experimental variability, whereby entire experimental groups appear to be “more lively” or “more dead”. The authors can use any cell viability assay of their choice, but if the resazurine assay is giving incredibly high variability, it is simply not a good one. That there is no mathematical significance in such cases is clear, but an average of 20% less signal and the clear trend at higher concentrations, is a quite substantial effect - at least in my opinion. This may be particularly important for longer ongoing experiments, such as the ones for RSV and SeV (48h knockdown, 24h virus growth). I thus had asked for knockdown kinetics and viability data, which was not addressed.

Note aside: Why does viability data in Figure 6b is all centered at 100% for NSC, while the figure in the rebuttal shows massive spread – figure legends read similarly? Given the similarity of these assays I would expect a similar appearance.

We appreciate the reviewer's observations regarding variability in viability measurements. As noted in the manuscript, the differences observed at low siRNA concentrations were not statistically significant, and our interpretations are based on statistical outcomes rather than apparent trends. While variability is inherent to biological assays, the resazurin assay remains a widely accepted and appropriate method for assessing cell viability. Therefore, we have reported and interpreted the data according to statistical significance, which is the most robust basis for our conclusions.

The difference observed in the original 6B vs the rebuttal was due to an error my team made in analysis which I addressed in the previous rebuttal with the following comment "This normalisation corrected for the seeming lack of variation initially shown for the control group. Now all replicates have been normalised to the mean control signal (with all average viabilities remaining unchanged), which corrects the control values and restores proper statistical comparability, see below)."

Minor:

Please note that there is a typo in the Figure legend of Fig 6b of the current manuscript: I suppose it should read, 10 uM NTC or decreasing doses of EHD2 siRNA.

This has been rectified to "A549 cells were transfected with the noted concentration of siEHD2 or 10nM siNSC for 48h..."

3. Lack of clear evidence of EHD2 and MYOF interaction:

This point was not addressed.

We have provided evidence of both EHD2 and MYOF occupying the same cellular space, associating with trafficking vRNP complexes. Additionally, as referenced in the manuscript on line 468, this interaction has already been proven and published by Doherty et al., 2008.